# Functional brain network reconfiguration during learning in a dynamic environment

Chang-Hao Kao [1✉], Ankit N. Khambhati[2], Danielle S. Bassett [3,4,5,6,7,8], Matthew R. Nassar [9,10], Joseph T. McGuire [11], Joshua I. Gold [12] & Joseph W. Kable[1✉]

When learning about dynamic and uncertain environments, people should update their beliefs most strongly when new evidence is most informative, such as when the environment undergoes a surprising change or existing beliefs are highly uncertain. Here we show that modulations of surprise and uncertainty are encoded in a particular, temporally dynamic pattern of whole-brain functional connectivity, and this encoding is enhanced in individuals that adapt their learning dynamics more appropriately in response to these factors. The key feature of this whole-brain pattern of functional connectivity is stronger connectivity, or functional integration, between the fronto-parietal and other functional systems. Our results provide new insights regarding the association between dynamic adjustments in learning and dynamic, large-scale changes in functional connectivity across the brain.

[1] Department of Psychology, University of Pennsylvania, Philadelphia, PA 19104, USA. [2] Department of Neurological Surgery, University of California, San Francisco, CA 94122, USA. [3] Department of Bioengineering, University of Pennsylvania, Philadelphia, PA 19104, USA. [4] Department of Neurology, University of Pennsylvania, Philadelphia, PA 19104, USA. [5] Department of Electrical & Systems Engineering, University of Pennsylvania, Philadelphia, PA 19104, USA. [6] Department of Physics & Astronomy, University of Pennsylvania, Philadelphia, PA 19104, USA. [7] Department of Psychiatry, University of Pennsylvania, Philadelphia, PA 19104, USA. [8] Santa Fe Institute, Santa Fe, NM 87501, USA. [9] Department of Neuroscience, Brown University, Providence, RI 02912, USA. [10] Robert J. and Nancy D. Carney Institute for Brain Science, Brown University, Providence, RI 02912, USA. [11] Department of Psychological and Brain Sciences, Boston University, Boston, MA 02215, USA. [12] Department of Neuroscience, University of Pennsylvania, Philadelphia, PA 19104, USA. ✉email: chakao@sas.upenn.edu; kable@psych.upenn.edu

Human decisions are guided by beliefs about current features of the environment. These beliefs often must be inferred from indirect and uncertain evidence. For example, deciding to go to a restaurant typically relies on a belief about its current quality, which can be inferred from past experiences at that restaurant. This inference process is particularly challenging in dynamic environments whose features can change unexpectedly (e.g., a new chef was just hired). In these environments, people tend to follow normative principles and update their beliefs dynamically and adaptively, such that beliefs are updated more strongly when existing beliefs are weak or irrelevant, and/or the new evidence is strong or surprising[1–3]. Recent studies have identified potential neural substrates of this adaptive belief-updating process, including univariate and multivariate activity patterns for uncertainty and surprise in several brain regions, including dorsomedial frontal cortex, anterior insula, lateral prefrontal cortex, and lateral parietal cortex[2,4–7]. The goal of the present study was to gain deeper insights into how these representations might interact dynamically to support adaptive belief updating.

We focused on how changes in belief updating relate to changes in functional connectivity between brain regions with task-relevant activity modulations. Functional connectivity reflects statistical dependencies between regional activity time series[8] and can form functional-connectivity networks that provide new perspectives on brain function[9–11]. Many recent studies of learning have focused on brain network reconfigurations occurring between naïve and well-learned phases in various domains such as motor, perceptual, category, spatial, or value learning[12–22]. In these cases, functional connectivity associated with the fronto-parietal system decreased gradually as learning progressed and this change in connectivity was associated with individual learning or performance[13,19,22]. In dynamic environments, however, people progressively learn the current state and then re-initialize learning once the state changes. Thus, we expected frequent reconfigurations in functional connectivity, as learning shifts between slower and faster updating in response to changes in uncertainty and surprise. In addition, although brain regions that encode uncertainty and surprise participate in multiple networks, including the fronto-parietal system, dorsal attention system, salience system, and memory system[2,4–7], based on previous network analyses of learning in stable environments we hypothesized that the fronto-parietal system would serve a particularly important role in network reconfiguration during learning in dynamic environments.

In the current study, we aimed to identify such frequent reconfigurations in functional connectivity during adaptive belief updating. A key to our approach was the use of an unsupervised machine-learning technique known as nonnegative matrix factorization (NMF)[23]. NMF decomposes the whole-brain network into subgraphs, which describe patterns of functional connectivity across the entire brain, and the time-dependent magnitude with which these subgraphs are expressed. Briefly, a subgraph is a weighted pattern of functional interactions that statistically recurs as the brain network evolves over time. We chose NMF because it provides two key advantages over other approaches to matrix factorization, such as principal components analysis (PCA) or independent components analysis (ICA)[24,25]. First, NMF yields a parts-based representation of the network, in which the individual components are strictly additive—a constraint that is not present in PCA and ICA. This important feature enables interpretation of the resulting subgraph and time-dependent expression coefficients on the basis of their positive distance from zero. Second, NMF does not enforce an orthogonality or independence constraint and, therefore, allows subgraphs to overlap in their structure. This property may more effectively model distinct subgraphs that may be jointly related via weak connections and better account for the flexibility of neural systems, such that one connection between regions can be involved in multiple systems or cognitive functions. Recently, NMF has been used to identify network dynamics during rest and task states[25,26] and to determine how these dynamics vary across development[24]. Here, we extend the use of this technique to examine changes in network dynamics linked to task variables and individual differences.

Our results show that that uncertainty and surprise, task variables that drive the adjustment of learning, are related to the temporal expression of specific patterns of functional connectivity (i.e., specific subgraphs). These specific patterns of functional connectivity prominently involve the fronto-parietal network. We also show that the dynamic modulation of these patterns of functional connectivity (i.e., subgraph expression) are associated with individual differences in learning.

## Results

**Belief updating is influenced by uncertainty and surprise.** Participants performed a predictive-inference task during functional magnetic resonance imaging (fMRI) (Fig. 1a). For this task, participants positioned a bucket to catch a bag that dropped from an occluded helicopter. The location of the bag was sampled with noise from a distribution centered on the location of the helicopter. The location of the helicopter usually remained stable but occasionally changed suddenly and unpredictably (with an average probability of change of 0.1 across trials). In addition, whether the bag (if caught) was rewarded or neutral was assigned randomly on each trial and indicated by color. This task challenged participants to form and update a belief about a latent variable (the location of the helicopter) based on noisy evidence (the location of dropped bags).

We previously described a theoretical model approximating the normative solution for this task[2]. This theoretical model takes the form of a delta-rule and approximates the Bayesian ideal observer. Beliefs ($B_{t+1}$) are updated based on the difference between the current outcome location ($X_t$) and the predicted location ($B_t$), with the extent of updating controlled by a learning rate ($\alpha_t$; Fig. 1b). Trial-by-trial learning rates are determined by two factors: (i) change-point probability (CPP), which is the probability that a change-point has happened and represents a form of belief surprise; and (ii) relative uncertainty (RU), which is the reducible uncertainty regarding the current state relative to the irreducible uncertainty that results from environmental noise and represents a form of belief uncertainty (Fig. 1c). Learning rates are higher when either CPP or RU is higher: $\alpha_t = \text{CPP} + (1 - \text{CPP})\text{RU}$.

We previously reported how participants' predictions were influenced by both normative and nonnormative factors and how these factors are encoded in univariate and multivariate activity[2,7]. Participants updated their beliefs more when the value of CPP or RU was higher, consistent with the normative model. Participants also updated their beliefs more when the outcome was rewarded, however, which is not a feature of the normative model. CPP, RU, and reward, as well as residual updating (belief updating not captured by CPP, RU, or reward), were all encoded in univariate and multivariate brain activity in distinct regions[2,7]. In the current study, we built on these previous findings and investigated how these factors, as well as individual differences in how these factors influence belief updating, are related to the dynamics of whole-brain functional connectivity.

**NMF identified ten subgraphs that varied over time.** We used NMF to decompose whole-brain functional connectivity over time into specific patterns of functional connectivity, called subgraphs, and quantified the expression of these patterns over time.

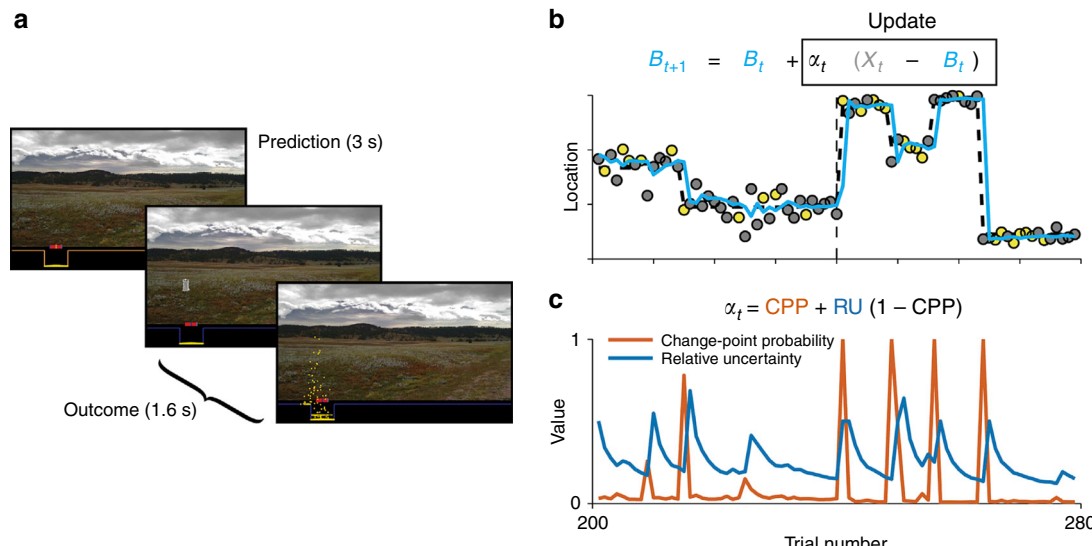

**Fig. 1 Overview of the task and theoretical model of belief updating (McGuire et al., 2014). a** Sequence of the task. At the start of each trial, participants predict where a bag will drop from an occluded helicopter by positioning a bucket on the screen. After participants submit their prediction, the bag drops and any rewarded coins that fall in the bucket are added to the participant's score. The location of the last prediction and the last bag drop are noted on the next trial. **b** An example sequence of trials. Each data point represents the location of a bag on each trial (yellow for rewarded coins, gray for neutral coins). The dashed line represents the true generative mean. The mean changes occasionally. The cyan line represents the prediction from a normative model of belief updating. The inset equation shows how the model updates beliefs ($B_t$ = belief, $X_t$ = observed outcome, $\alpha_t$ = learning rate on trial $t$). The vertical dashed line represents the boundary of the noise conditions: high-noise (left) and low-noise condition (right). Noise refers to the variance of the generative distribution. **c** Two learning components from the normative model. Change-point probability (CPP) reflects the likelihood that a change-point happens, which is increased when there is an unexpectedly large prediction error. Relative uncertainty (RU) reflects the uncertainty about the generative mean relative to the environmental noise, which is increased after high CPP trials and decays slowly as more precise estimates of the generative mean are possible. The inset formula shows how CPP and RU contribute to single trial estimates of learning rates.

To perform NMF, we first defined regions of interest (ROIs) based on a previously defined parcellation[27] (Fig. 2a) and extracted blood-oxygenation-level-dependent (BOLD) time series for each ROI (Fig. 2b). For every pair of ROIs, we calculated the Pearson correlation coefficient between the BOLD time series in 10-TR (25 s) time windows, offset by 2 TRs for each time step (and thus 80% overlap between consecutive time windows). This procedure thus yielded a matrix whose entries represented time-dependent changes in the strengths of these pairwise correlations in the brain during the task. We unfolded each time window from this correlation matrix (Fig. 2c) into a one-column vector, and then concatenated these vectors from all time windows and all participants (Fig. 2d). As required for NMF, we transformed the resulting matrix to have strictly nonnegative values: we duplicated the full matrix, set all negative values to zero in the first copy, and set all positive values to zero in the second copy before multiplying all remaining values by negative one. Thus, we divided the final full data matrix into two-halves, with one-half containing the positive correlation coefficients (zero if the coefficient was negative) and one-half containing the absolute values of the negative correlation coefficients (zero if the coefficient was positive)[26]. This procedure ensured that our approach did not give undue preference to either positive or negative functional connectivity, and that subgraphs were identified based on both positive and negative functional connectivity.

We applied NMF to this matrix (**A**) to identify functional subgraphs and their expression over time. Specifically, we decomposed the full data matrix into a subgraph matrix **W** and an expression matrix **H** (Fig. 2d). The columns of **W** represent different subgraphs and the rows represent different edges (i.e., pairs of regions), with the value in each cell representing the strength of that edge (i.e., the functional connectivity strength for that pair of regions) for that subgraph. The rows of **H** represent different subgraphs, and the columns represent time windows, with the value in each cell representing the degree of expression of that subgraph in that time window. We implemented NMF by minimizing the residual error ($||\mathbf{A} - \mathbf{WH}||_F^2$) via three parameters: (i) the number of subgraphs ($k$), (ii) the subgraph regularization ($\alpha$), and (iii) the expression sparsity ($\beta$) (Supplementary Fig. 1).

Using NMF, we identified ten subgraphs, which reflected patterns of functional connectivity strengths across every pair of regions in the brain, as well as the expression of these subgraphs over time. The full description of each subgraph specifies the edge strength between every pair of ROIs, corresponding to a 247 × 247 matrix. We calculated a simpler summary description that specifies the edge strength between every pair of functional systems in the previously defined parcellation, corresponding to a 13 × 13 matrix[27]. Edges between ROIs were categorized according to the functional system of each ROI. To estimate the diagonal entries in the system-by-system matrix, we averaged the weights of all edges connecting two ROIs within a given system (Fig. 3a). To estimate the off-diagonal entries of the system-by-system matrix, we averaged the weights of all edges linking an ROI in one system with an ROI in another system. In line with common parlance, we refer to the edges within the same system as within-system edges, whereas we refer to the edges between two different systems as between-system edges. For presentation, we ordered and numbered the ten subgraphs according to the strength of within-system edges relative to that of between-system edges (Fig. 3b, Supplementary Fig. 2a–c). Finally, we thresholded the system-by-system matrix to show only edges that passed a permutation test ($p < 0.05$ after the Bonferroni correction for multiple comparisons; see Methods). The full data matrix on

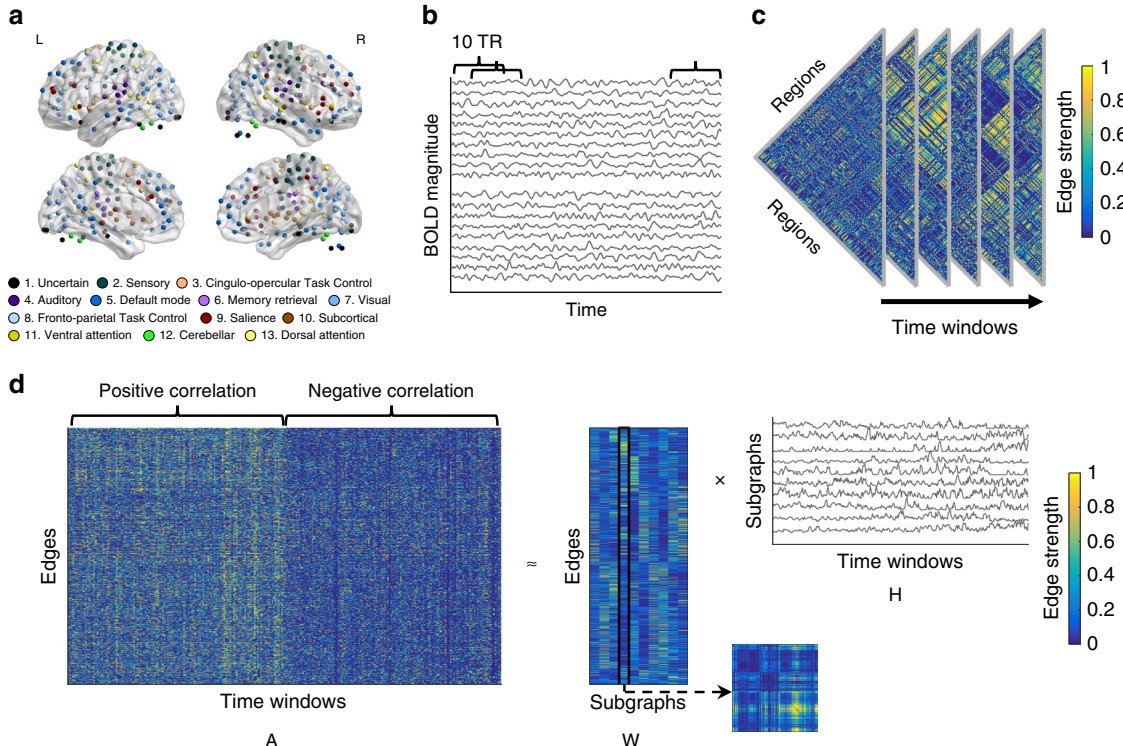

**Fig. 2 Schematic overview of the method. a** Regions of interest (ROIs). Functional MRI BOLD signals were extracted from spherical ROIs based on the previously defined parcellation[27]. We only kept 247 ROIs that had usable data from all subjects. Each ROI can be assigned to one of 13 putative functional systems. The brain figure was visualized by the BrainNet Viewer[42] under the Creative Commons Attribution (CC BY) license (https://creativecommons. org/licenses/by/4.0/). **b** An example of Pearson correlation coefficients calculated between regional BOLD time series over the course of the experiment. Each BOLD time series was divided into 10-TR (25 s) time windows, and consecutive time windows were placed every 2 TRs leading to 80% overlap between consecutive time windows. Pairwise Pearson correlation coefficients were calculated between ROI time series in each time window. **c** An example of edge strength over time. In each time window, there were $247*(247 − 1)/2$ edges. **d** Nonnegative matrix factorization (NMF). In each time window, the matrix of edge strengths was unfolded into one column. Then, edges from all time windows in all participants were concatenated into a single matrix. Each row in the full data matrix contained an edge (pairwise correlation coefficients between BOLD time series from two ROIs) and each column contained a time window (across all scans and participants). Correlation values in this matrix were strictly non-negative; the full data matrix was divided into two halves, with one half containing the positive pairwise correlation coefficients (zero if the correlation coefficient was negative) and one half containing the absolute values of negative pairwise correlation coefficients (zero if the correlation coefficient was positive). Thus, subgraphs were identified based on both the similarity of positive functional connectivity and the similarity of negative functional connectivity together. Then, NMF was applied to decompose the concatenated matrix into a matrix **W**, which encoded the strengths of edges for each subgraph, and a matrix **H**, which encoded the time-dependent expression of each subgraph. For example, the strength of edges of the fourth subgraph (the fourth column in the matrix **W**) can be folded into a squared matrix, reflecting the edge strength between every pair of ROIs.

which we performed NMF was divided into two-halves, with the first half corresponding to positive functional connectivity and the second half corresponding to negative functional connectivity. The expression matrix **H** was therefore also divided into two-halves, with the first half corresponding to positive expression over time and the second half corresponding to negative expression over time. Positive and negative expression coefficients were highly negatively correlated with each other across time for all the subgraphs (all $r < −0.61$, all $p < 0.001$). For the analyses of subgraph expression below, we thus constructed a measure of relative subgraph expression by subtracting the negative expression from the positive expression at each time point[26]. Across subgraphs, the average relative expression across time was strongly correlated with the relative strength of within- versus between-system edges (Supplementary Fig. 2d–f). That is, higher within-system strength was associated with greater relative expression of the subgraph.

**Normative factors modulated subgraph expression**. We investigated how CPP, RU, reward, and residual updating influenced the temporal expression of each subgraph. We identified a

particularly strong relationship between the normative factors (CPP, RU, and the residuals that reflected the participants' subjective estimates of those variables) and subgraph 4, whose strongest edges were in the fronto-parietal task-control system, followed by the memory retrieval, salience and dorsal-attention systems (Fig. 4a, b). Specifically, we used multiple regression to estimate the trial-by-trial relationship between these four factors and the relative expression strength of each subgraph. For each subgraph, regression coefficients were fitted separately for each participant and were tested at the group level using $t$ tests (Supplementary Fig. 3). Among the ten subgraphs, these four factors explained the most variance in the time-dependent relative expression of subgraph 4 (Supplementary Fig. 4), in each case showing positive modulations (CPP: mean ± SEM = 0.202 ± 0.053, $t_{31} = 3.78$, $p < 0.001$; RU: 0.392 ± 0.077, $t_{31} = 5.11$, $p < 0.001$; residual updating: 0.177 ± 0.079, $t_{31} = 2.23$, $p = 0.033$; Fig. 4c). We also evaluated the influence of head motion by including motion, as indexed as the relative root-mean-square of the six motion parameters, in the regression model. Motion was not significant ($p = 0.29$) and the effects of CPP, RU, and residual updating remained significant and of similar effect size.

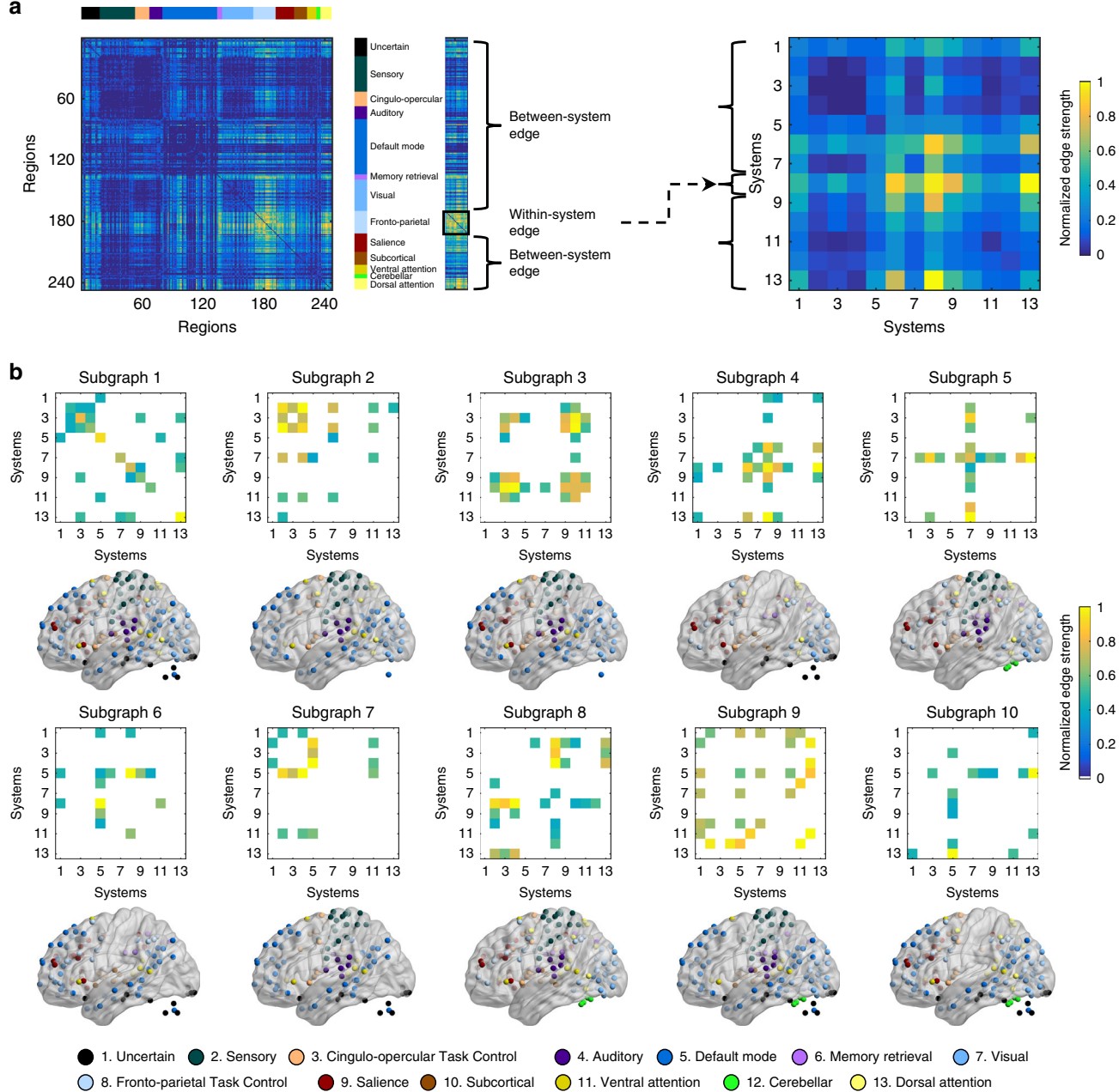

**Fig. 3 Patterns of connectivity in subgraphs. a** Converting edges between nodes into edges between systems. First, the edges of each subgraph can be folded into a square matrix, representing the edges between every pair of nodes (ROIs). Then, based on the 13 putative functional systems reported by Power et al. (2011), we categorized each edge according to the system(s) to which the two nodes (ROIs) belonged. We calculated the mean strength of edges linking a node in one system to a node in another system, and refer to that value as the between-system edge. Similarly, we calculated the mean strength of edges linking two nodes that both belong to the same system and refer to that value as the within-system edge. Edges between nodes and edges between systems were normalized into the scale between 0 and 1. **b** Edges between systems in the ten subgraphs identified by NMF. We show only significant edges ($p < 0.05$ after the Bonferroni correction for multiple comparisons). For each subgraph, the top matrix shows the significant edges in that subgraph within or between systems. For example, Subgraph 1 has high edge strengths along the diagonal; thus, this subgraph describes functional connectivity that lies predominantly within functional systems. In contrast, subgraph 5 has high edge strengths along a single row and column, corresponding to the visual system; thus, this subgraph describes functional connectivity between the visual system and all other systems. Subgraphs varied in the degree to which they represent interactions within the same system (e.g., subgraph 1) versus interactions between different systems (e.g., subgraph 10). All nodes from systems involved in significant edges are shown on the brain below by the BrainNet Viewer[42] under the Creative Commons Attribution (CC BY) license (https://creativecommons.org/licenses/by/4.0/).

Although CPP or RU also modulated the relative expression of some other subgraphs (e.g., subgraphs 1, 3, and 7; Supplementary Fig. 3), below we focus on subgraph 4 for several reasons. First, the four factors we investigated explained more variance in the time-dependent relative expression of subgraph 4 than that of any other subgraph. Second, only on subgraph 4 were the effects of CPP and RU strong enough to survive correction for multiple comparisons across ten subgraphs. Third, only on subgraph 4 were the effects of CPP and RU robustly shown across analyses using different sized time windows.

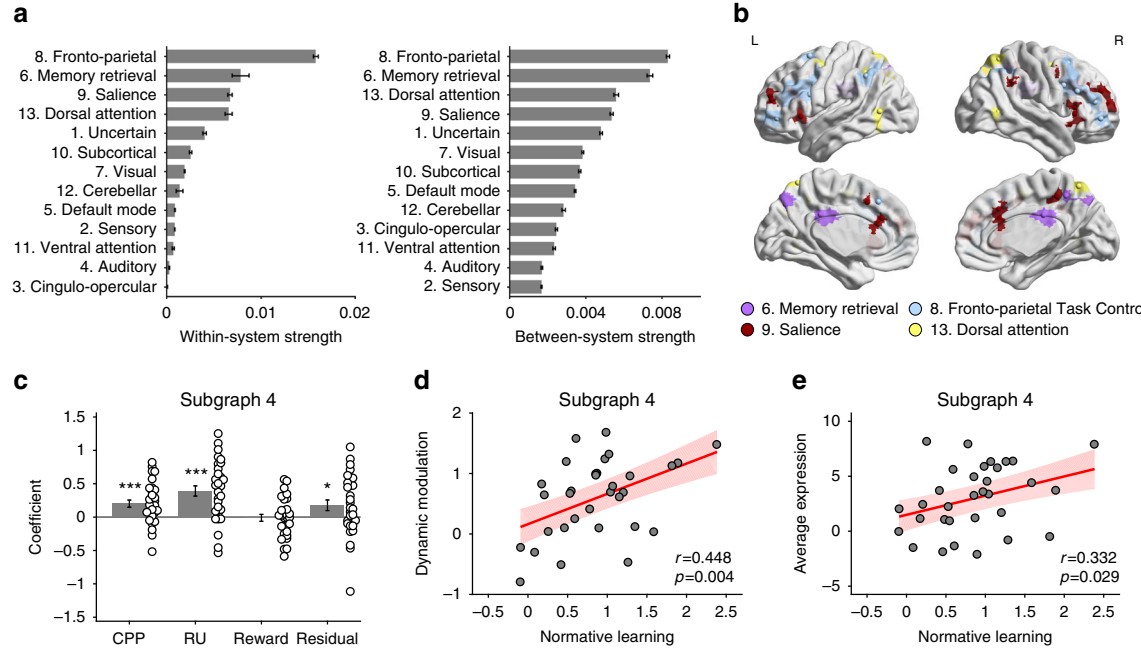

**Fig. 4 Temporal expression of subgraph 4 was related to task factors and individual differences. a** Summary of the pattern of connectivity in subgraph 4. We summarized the pattern of connectivity as within-system strength (which is the value in the diagonal) and between-system strength (which is the average of values in the off-diagonal) for each system. The fronto-parietal system as well as three other systems (memory retrieval, salience, and dorsal attention) showed the strongest contributions to this subgraph in terms of both within-system and between-system strength. The 95% confidence interval of each system was estimated by boostrapping 10,000 times on the edges of that system. **b** Nodes for the top four systems with strong within-system and between-system strength. We showed the nodes of fronto-parietal system, memory retrieval system, salience system and dorsal attention system on the brain by the BrainNet Viewer[42] under the Creative Commons Attribution (CC BY) license (https://creativecommons.org/licenses/by/4.0/). **c** Modulation of temporal expression of subgraph 4 by task factors. A regression model that included CPP, RU, reward, and residual updating as predictors of temporal relative expression (calculated by subtracting negative expression from positive expression) of subgraph 4 was fitted for each participant, and coefficients were tested on the group level by t tests. The results showed positive effects of CPP, RU, and residual updating. Each point represents one participant. Error bars represent one SEM. (*$p < 0.05$, ***$p < 0.001$) **d** The relationship between individual normative learning and the dynamic modulation of subgraph 4 expression by normative factors. This dynamic modulation was indexed as the sum of the coefficients of CPP and RU in (**c**), and represents the extent to which trial-by-trial expression was influenced by the two normative learning factors. There was a significant positive correlation across participants. Each point represents one participant. The red line represents the regression line and the shaded area represents the 95% confidence interval. **e** The relationship between individual normative learning and average relative expression of subgraph 4. There was a significant positive correlation across participants. Each point represents one participant. The red line represents the regression line and the shaded area represents the 95% confidence interval. Source data of **c**–**e** are provided as a Source Data file.

**Individual differences associated with subgraph expression.** The expression of subgraph 4 was not only modulated by task factors that drive normative learning, but also varied across subjects in a manner that reflected individual differences in normative learning. As an index of normative learning, we estimated the influence of CPP and RU on trial-by-trial belief updates using multiple regression and took the sum of the regression coefficients of CPP ($\beta_2$ in Eq. (6)) and RU ($\beta_3$ in Eq. (6)) for each participant[2]. This sum reflected how much each individual updated their beliefs in response to normative factors. We examined the relationship between individual differences in this normative belief-updating metric and two aspects of subgraph expression.

First, we examined the relationship between normative belief updating and the dynamic modulation of subgraph expression by normative factors (Supplementary Fig. 5). As an index of the dynamic modulation of subgraph expression by normative factors, we used the sum of the regression coefficients of CPP and RU on relative expression from the analyses above (Supplementary Fig. 3). We found a positive correlation between the dynamic modulation of subgraph 4 expression by normative factors and normative belief updating across participants

($r = 0.448$, $p = 0.004$; Fig. 4d). Second, We also found a positive correlation between the average relative expression of subgraph 4 and normative belief updating across participants ($r = 0.332$, $p = 0.029$; Fig. 4e; Supplementary Fig. 6). These effects were still significant when we controlled for the influence of motion on dynamic modulation or average relative expression, whereas the effects of motion itself were not significant (all $p > 0.31$). These two results show that participants with the highest average relative expression of subgraph 4, and for whom the normative factors account for the most variance in the relative expression of subgraph 4 across time, tended to update their beliefs in a manner more consistent with the normative model than the other subjects.

**Contribution of specific edges to the identified effects.** Subgraph 4 describes both within- and between-system functional connectivity for multiple functional systems (Figs. 3b and 4a, b; Supplementary Fig. 2a–c). We next examined the contribution of specific edges (i.e., functional connectivity between specific pairs of brain regions) within subgraph 4 to the task and individual difference effects we observed for that subgraph.

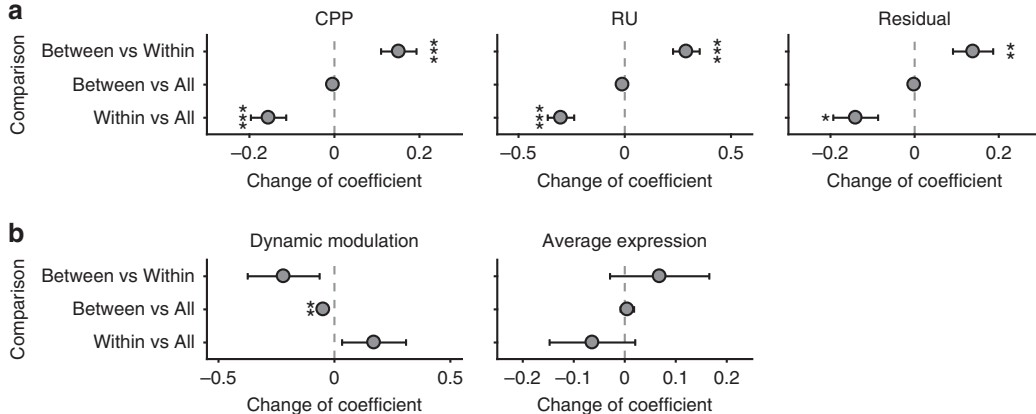

**Fig. 5 The contribution of between-system and within-system edges to effects of task factors and individual differences on subgraph 4 expression. a** The contribution of between-system and within-system edges to the effect of task factors on temporal relative expression of subgraph 4. To determine the relative contribution of between- and within-system edges on time-dependent subgraph 4 expression, we performed three comparisons on the effects estimated by different types of edges using $t$ tests: within-system edges only (Within), between-system edges only (Between) and all edges (All). First, removing between-system edges (Within versus All) decreased the effect of CPP, RU and residual updating. Second, in contrast, after removing within-system edges (Between versus All), there was no significant change in these coefficients. Third, we directly compared the effects contributed from between-system edges only and from within-system edges only (Between versus Within). For between-system edges, there were stronger positive effects for CPP, RU, and residual updating. Error bars represent one SEM. (*$p < 0.05$, **$p < 0.01$, ***$p < 0.001$). **b** The contribution of between-system and within-system edges to the relationship between normative learning and dynamic modulation and average expression of subgraph 4. We performed the same three comparisons to determine the relative contribution of between- and within-system edges for each relationship with individual differences. For the effect of dynamic modulation, removing within-system edges (Between versus All) decreased the correlation coefficient. This correlation coefficient was also larger for within-system edges only than between-system edges only, but this effect was not statistically significant. For the effect of average expression, removing between-system edges (Within versus All) decreased the correlation coefficient, and the correlation coefficient was larger for between-system edges only than within-system edges only, though neither of these effects were statistically significant. Source data are provided as a Source Data file. Error bars represent one SEM. (**$p < 0.01$).

The task-related modulations of subgraph 4 involved primarily between-system, not within-system, functional connectivity. Specifically, we re-estimated the effects of CPP, RU, reward, and residual updating on the relative expression of subgraph 4 using only within-system edges (i.e., only the diagonal cells of the system-by-system matrix in Fig. 3b; "Within") or only between-system edges (i.e., only the off-diagonal cells of the system-by-system matrix in Fig. 3b; "Between"). We compared these effects to our previous estimates using all edges (Fig. 5a; "All") through $t$ tests. Removing the between-system edges (Within versus All) reduced the size of the estimated effects of CPP (mean ± SEM = $-0.155 ± 0.042$, $t_{31} = -3.73$, $p < 0.001$), RU ($-0.300 ± 0.062$, $t_{31} = -4.82$, $p < 0.001$), and residual updating ($-0.140 ± 0.053$, $t_{31} = -2.63$, $p = 0.013$). In contrast, removing the within-system edges (Between versus All) led to no reliable changes in these effects (all $p > 0.21$). Further, in a direct comparison of the reduced subgraphs with only within- or between-system edges, the effects estimated with between-system edges only were stronger for CPP ($0.151 ± 0.042$, $t_{31} = 3.63$, $p < 0.001$), RU ($0.290 ± 0.063$, $t_{31} = 4.63$, $p < 0.001$), and residual updating ($0.139 ± 0.048$, $t_{31} = 2.91$, $p = 0.007$).

The contributions of within- and between-system functional connectivity to the individual difference effects of subgraph 4 were less clear. For the relationship between individual differences in normative learning and average relative expression, the pattern across comparisons was similar to that observed for task effects (Fig. 5b), which would indicate a greater contribution of between-system edges, but none of the comparisons were statistically significant. In contrast, for the relationship between individual differences in normative learning and the dynamic modulation of subgraph 4, within-system edges appeared to be more important, as removing the within-system edges (Between versus All) reduced this correlation (difference = 0.048, $p = 0.006$; Fig. 5b).

Supplementary analyses identified contributions of specific functional systems (i.e., one row/column from the system-by-system matrix in Fig. 3b; Supplementary Fig. 7) and of specific system-by-system edges (i.e., one cell from the system-by-system matrix in Fig. 3b; Supplementary Fig. 8) to the task and individual difference effects on subgraph 4.

**Robust effects across different sized time windows**. To determine the sensitivity of our results to the size of this time window, we repeated the entire procedure using shorter (8-TR/20 s window with 6-TR/15 s overlap; Supplementary Figs. 9–12) or longer (12-TR/30 s window with 10-TR/25 s overlap; Supplementary Figs. 13–16) time windows. That is, we shorten or lengthen the time window by the interval of one trial (~5 s). With both shorter and longer time windows, we identified ten subgraphs. There was a high degree of similarity between the ten subgraphs identified in the main analysis and those identified using either shorter (edges between nodes: all $r > 0.81$; edges between systems: all $r > 0.80$) or longer (edges between nodes: all $r > 0.98$; edges between systems: all $r > 0.98$) time windows. With longer time windows, the relative expression of subgraph 4 still showed the same relationship to task factors (CPP and RU) and to individual differences in normative learning; with shorter time windows, these effects were also present but weaker.

**Relationship between regional activity and connectivity**. In our previous report, we described how CPP, RU, reward, and residual updating influenced univariate brain activity. In a final set of analyses, we examined the relationship between these previously reported univariate effects and the changes in dynamic functional connectivity we identified above.

The brain regions that were most strongly represented in subgraph 4 overlapped spatially with the brain regions whose

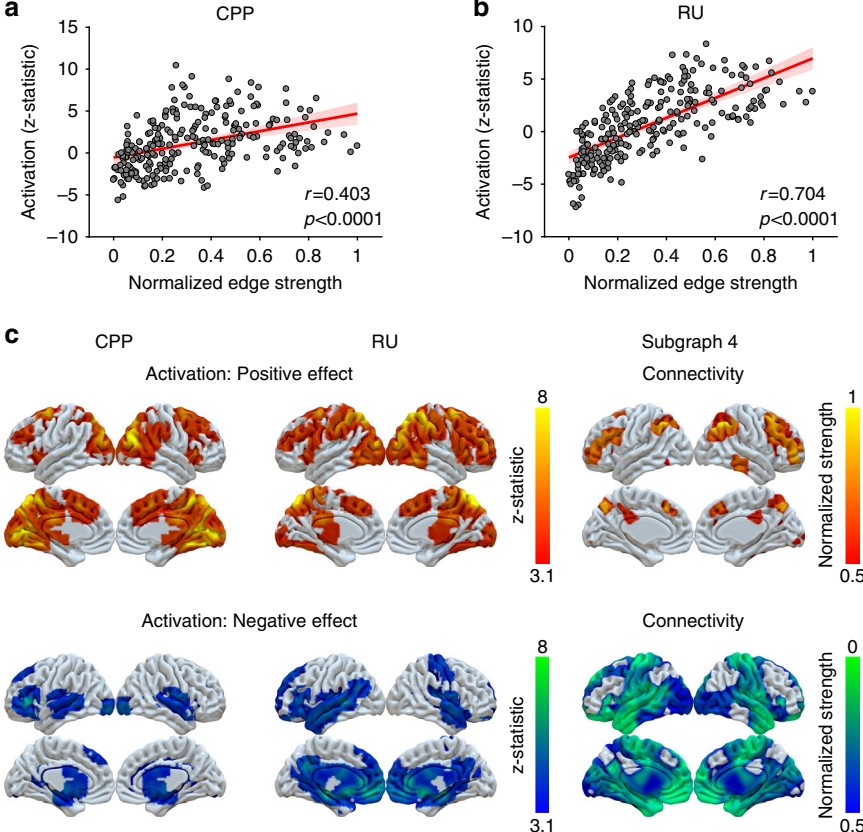

**Fig. 6 Relationship between edge strength of subgraph 4 and univariate task activations. a** Relationship between the activation for CPP and the edge strength of subgraph 4. We calculated the Pearson correlation coefficient between the z-statistic for CPP from McGuire et al. (2014) and the edge strength across nodes in subgraph 4. Each data point represents an ROI. The edge strength for each ROI was calculated as the column sum of that ROI's edges to other ROIs, reflecting the summed interactions between that ROI and all others. The edges were normalized into the scale between 0 and 1. A significantly positive correlation was observed. The red line represents the regression line and the shaded area represents the 95% confidence interval. **b** Relationship between the activation for RU and the edge strength of subgraph 4. We observed a significant positive correlation between the z-statistic for RU from McGuire et al. (2014) and the edge strength across nodes in subgraph 4. The red line represents the regression line and the shaded area represents the 95% confidence interval. Source data of **a**, **b** are provided as a Source Data file. **c** Whole-brain thresholded activation maps for CPP and RU from McGuire et. al (2014) and whole-brain maps for edge strength of subgraph 4 in the current study.

activity was modulated reliably by CPP and RU in our previous report. As a measure of a region's involvement in subgraph 4, for each ROI, we calculated the mean strength of every edge between that ROI and all other ROIs in subgraph 4, and normalized these mean values between 0 and 1. We then related this metric to activation from our previous study[2], as measured by the z-statistic of the modulation effect of CPP or RU. This z-statistic indicated the effect size of change of univariate activity in response to CPP or RU across participants. Across all ROIs, there was a positive correlation between edge strength in subgraph 4 and activation for CPP ($r = 0.403$, $p < 0.0001$; Fig. 6a) and activation for RU ($r = 0.704$, $p < 0.0001$; Fig. 6b). The Surf Ice software (https://www.nitrc.org/projects/surfice) was used to show the map of normalized mean edge strengths for subgraph 4 alongside the thresholded activation maps for CPP and RU (Fig. 6c). Regions with stronger edge strength in subgraph 4, such as the insula, dorsomedial frontal cortex, dorsolateral prefrontal cortex, posterior parietal cortex, and occipital cortex, also tended to show stronger increases in activation with increases in CPP and RU.

In addition to these strong associations between univariate brain activation and edge strength, effects beyond those captured by univariate task activity also contributed to our dynamic functional connectivity results. To demonstrate this, we estimated functional connectivity from time-series that only contained task-modulated univariate activity, performed NMF on this matrix,

and repeated all of our main analyses (Supplementary Figs. 17–20). This analysis again identified a subgraph 4 whose strongest edges were in the fronto-parietal system, but it did not recapitulate all of the relationships between subgraph 4 expression and task factors and individual differences seen in our main analyses. These results implied that the dynamic functional connectivity patterns identified in our main analyses reflect a mixture of coordinated activity across regions (which can be captured by univariate analyses) and other statistical dependencies across regions that require network-based analyses.

## Discussion
We identified a pattern of dynamic functional brain connectivity in human subjects performing a predictive-inference task. This pattern was expressed most strongly during times that demanded faster belief updating and was enhanced in individuals who most effectively used adaptive belief updating to perform the task. To identify this pattern, we used NMF, an unsupervised machine-learning technique that decomposes the full matrix of time-dependent functional connectivity into subgraphs (patterns of functional connectivity), and the time-dependent magnitude of these subgraphs. Among the subgraphs we identified in our data, the expression of one subgraph in particular was modulated reliably by three trial-by-trial factors that influenced the degree of

behavioral belief updating: CPP (surprise), RU (uncertainty), and residual updating (updating unaccounted for by surprise or uncertainty). Notably, CPP and RU are factors that normatively promote greater belief updating, scaling the degree to which past observations are discounted relative to the most recent evidence. Residual updating likely captures, at least in part, deviations between the objective values of CPP and RU in the normative model and the individual's subjective estimates of those factors. Thus, the expression of this subgraph reflects not only normative factors that should influence belief updating but also likely fluctuations in subjective estimates of those factors. In addition to being modulated by these trial-by-trial task factors, expression of this subgraph also varied across individuals in a manner associated with individual differences in belief updating. Participants who tended to update their beliefs in a more normative manner—that is, with a stronger influence of surprise (CPP) and uncertainty (RU)—showed stronger dynamic modulation of the expression of this subgraph by normative factors and showed stronger average expression of this subgraph.

The subgraph modulated by surprise and uncertainty included interactions between multiple functional systems, most prominently the fronto-parietal task control, memory retrieval, salience, and dorsal attention systems (Figs. 3b and 4a). These systems, include multiple regions in the anterior insula, dorsolateral and dorsomedial frontal cortex, and lateral and medial parietal cortex (Figs. 4b and 6c). These regions showed a large degree of overlap with areas that we have previously shown to have increased univariate activation in response to both surprise and uncertainty (in this same dataset; Fig. 6)[2]. A smaller subset of these regions, including parts of the dorsomedial frontal cortex, anterior insula, inferior frontal cortex, posterior cingulate cortex, and posterior parietal cortex, was modulated not only by both normative (surprise and uncertainty) factors, but also by a non-normative one (reward). This smaller subset includes regions that participate in the fronto-parietal task-control, memory retrieval, salience, and dorsal attention systems.

Previously, we also reported regions whose univariate activity was modulated by either surprise or uncertainty alone. Surprise was associated selectively with activation in occipital cortex, and uncertainty was associated selectively with activation in anterior prefrontal and parietal cortex[2]. We similarly have reported multivariate activation patterns that were associated selectively with either surprise or uncertainty alone[7]. In the current study, we identified a key pattern of functional connectivity that was modulated by both surprise and uncertainty, but we did not identify any other pattern that was modulated reliably by either surprise or uncertainty alone. One possible explanation for this lack of a positive result was our need to use relatively long time windows (25 s, corresponding to 4–6 trials) in order to obtain reliable estimates of functional connectivity. These time windows likely included both the surprise elicited by change-points and the uncertainty that follows. Thus, functional connectivity related to surprise and uncertainty may have been difficult to dissociate temporally in our task and analysis design. Using a task that can temporally separate the tracking of surprise and uncertainty[28] might enable the identification of distinct patterns of functional connectivity for each factor.

The identified pattern of whole-brain functional connectivity was also expressed across individuals in a manner that varied with the degree to which they updated their beliefs more in line with the normative model. Thus, individual differences in learning were also reflected in features of individual functional connectomes. In our previous study, we noted a relationship between individual differences in normative learning and the degree to which activity in dorsomedial frontal cortex and anterior insula was modulated by normative factors (surprise and uncertainty)[2].

Here, we showed that normative learning was also associated with how functional connectivity was modulated dynamically by the same normative factors. These new findings add to previous work showing that brain network dynamics can reflect individual differences in learning in various domains[12,13,15,19,22]. Potentially, these differences in individual functional connectomes during learning could reflect individual differences in resting-state (task-independent) functional connectivity[29], which merits further study.

Functional connectivity captures many different kinds of statistical dependencies between brain regions, including those that result from task-driven co-activation. The strong association between neural activation and functional connectivity during periods of surprise and uncertainty in our results (Fig. 6), as well as previous studies in other domains[13,15,17,19,21,22], raises the possibility that the increases in functional connectivity between brain regions might have arisen because these regions became more tightly synchronized to external task events, without necessarily any increase in communication between them. To refute this possibility, we repeated our analyses on the predicted BOLD time series from univariate GLMs. These predicted time series, which contain only task-driven statistical dependencies between brain regions, could not recapitulate all of the effects that we observed in our actual BOLD time series. Specifically, we found modulations by task (e.g., the modulation of subgraph expression by surprise and residual updating) and individual differences (e.g., the relationship between individual differences in normative learning and the dynamic modulation of subgraph expression by normative factors) that were apparent only in the full, original functional connectivity matrices. Thus, these effects appear to include neural communications that do not simply reflect task-driven co-activation. Even though the changes in functional connectivity that we describe may reflect a mixture of task-driven and endogenous dynamics, the network analysis provides an important higher-level, reduced-dimensionality description of these changes.

A key feature of the brain-wide pattern of functional connectivity that we identified was connectivity involving the fronto-parietal task-control system. We characterized the complex pattern of functional connectivity in the learning-related subgraph by summarizing the connectivity according the putative functional system of each region[27]. Among all the functional systems, the largest proportion of connectivity in the learning-related subgraph involved the fronto-parietal system. Connectivity associated with the fronto-parietal system has been shown to increase at the beginning of learning and decrease toward the later phases of learning[13,19,22]. Our result extends this finding by showing that fronto-parietal functional connectivity is modulated dynamically in a trial-by-trial manner according to the need for new learning. That is, the pattern of functional connectivity captured by the learning-related subgraph increased after surprising task changes and then decreased gradually as more information was gained about the current state. The fronto-parietal system is thought of as a control system that is involved in flexible adjustments of behavior[30,31]. In particular, connectivity between the fronto-parietal network and other systems has been shown to change in response to different task requirements[32]. This type of flexible control is critical for learning in a dynamic environment, a context in which people should adjust their degree of belief updating in a context-dependent manner[1,4].

Although the learning-related subgraph was also characterized by a balanced strength of within-system connectivity and between-system connectivity, the critical features that changed in response to task dynamics involved primarily between-system connectivity. This result implies that faster learning was associated with a greater degree of integration between different

functional systems. Several previous studies have shown that complex cognitive tasks are associated with more integration between systems[33–36]. Other work has shown that as a task becomes more practiced over time, the interaction between systems decreased while the connections within systems remained strong[13]. Here, we demonstrated changes in integration on a fast time scale, as task demands varied from trial to trial. Integration between systems was greater during periods of the task when surprise or uncertainty was high, and therefore there was a need to update one's beliefs and base them more on the current evidence than on expectations developed from past experience.

In this study, we provided a network-based perspective on the neural substrates of learning in dynamic and uncertain environments. In such environments, people should flexibly adjust between slow and fast learning: beliefs should be updated more strongly when new evidence is most informative, such as when the environment undergoes a surprising change or beliefs are highly uncertain. Here, we identified a specific brain-wide pattern of functional connectivity (subgraph) that fluctuated dynamically with changes in surprise and uncertainty. The dynamics and expression of this pattern of functional connectivity also varied across individuals in a manner that reflected differences in learning. This pattern was expressed more strongly and was more strongly modulated by surprise and uncertainty in people who updated their beliefs in a more normative manner, with a stronger influence of surprise and uncertainty. The most important aspect of this learning-related pattern of functional connectivity is functional integration between the fronto-parietal and other functional systems. These results establish a novel link between dynamics adjustments in learning and dynamic, whole-brain changes in functional connectivity.

## Methods

**Participants**. The dataset has been described in our previous reports[2]. Thirty-two individuals participated in the fMRI experiment: 17 females, mean age = 22.4 years (SD = 3.0; range: 18–30). Human subject protocols were approved by the Internal Review Board in University of Pennsylvania. All participants provided informed consent before the experiment.

**Task**. Each participant completed four 120-trial runs during functional magnetic resonance imaging. In each run, participants performed a predictive-inference task (Fig. 1a). On each trial, participants made a prediction about where the next bag would be dropped from an occluded helicopter by positioning a bucket along the horizontal axis (0–300) of the screen. The location of the bag was sampled from a Gaussian distribution with a mean (the location of the helicopter) and a standard deviation (noise). The standard deviation was high (SD = 25) or low (SD = 10) in different runs. The location of the helicopter usually remained stable but it changed occasionally. The probability of change was zero for the first three trials after a change and 0.125 for the following trials. When the location changed, the new location was sampled from a uniform distribution. Correctly predicting the location of the bag resulted in coins landing in the bucket. These coins either had positive or neutral value depending on their color, which was randomly assigned for each trial.

**Behavior model**. We applied the same normative model described in our previous study[2]. An approximation to the ideal observer solution to this task updates beliefs according to a delta learning rule (Fig. 1b)

$$\delta_t = X_t - B_t, \tag{1}$$

$$B_{t+1} = B_t + \alpha_t \times \delta_t, \tag{2}$$

where $\delta_t$ is the prediction error, which is the difference between the observed outcome (bag drop location, $X_t$) and the prediction (bucket location, $B_t$). Beliefs are updated in proportion to the prediction error, and this proportion is determined by $\alpha_t$, the learning rate. The learning rate is adjusted adaptively on each trial according to two normative factors (Fig. 1c)

$$\alpha_t = \Omega_t + (1 - \Omega_t) \times \tau_t, \tag{3}$$

where $\Omega_t$ is the CPP and $\tau_t$ is the RU. The learning rate, CPP and RU are all constrained to be between zero and one, and the learning rate increases when either CPP or RU is high. CPP reflects the likelihood that a change-point has happened[1,2]

$$\Omega_t = \frac{U(X_t|0, 300)H}{U(X_t|0, 300)H + N(X_t|B_t, \sigma_t^2)(1 - H)}, \tag{4}$$

where $U(X_t|0, 300)$ indicates the probability of $X_t$ from a uniform distribution between 0 and 300, $N(X_t|B_t, \sigma_t^2)$ indicates the probability of $X_t$ from a Gaussian distribution with mean of $B_t$ and variance of $\sigma_t^2$, $\sigma_t^2$ is the variance of predictive distribution of the bag location, and $H$ is the average probability of change (0.1) across trials.

RU reflects the uncertainty about the current location of the helicopter relative to the amount of noise in the environment[2]

$$\tau_{t+1} = \frac{\Omega_t \sigma_N^2 + (1 - \Omega_t)\tau_t \sigma_N^2 + \Omega_t(1 - \Omega_t)(\delta_t(1 - \tau_t))^2}{\Omega_t \sigma_N^2 + (1 - \Omega_t)\tau_t \sigma_N^2 + \Omega_t(1 - \Omega_t)(\delta_t(1 - \tau_t))^2 + \sigma_N^2}, \tag{5}$$

where $\sigma_N^2$ is the variance of outcome distribution used to generate the location of bag. There are three terms present in both the numerator and denominator. The first term is the variance of the helicopter distribution conditional on a change-point while the second term is the variance of the helicopter distribution conditional on no change-point. The third term reflects the variance due to the difference in mean between the two conditional distributions. The three terms together capture the uncertainty about the location of the helicopter.

Figure 1c shows an example of the dynamics of CPP and RU. CPP increases when there is an unexpectedly large prediction error. RU increases after CPP increases and decays slowly as more precise estimates of the helicopter location are possible.

As in our previous study, a regression model was applied to investigate how the factors in this normative model, as well as other aspects of the task, influenced participants' belief updates. We regressed trial-by-trial updates ($B_{t+1} - B_t$) against the prediction error ($\delta_t$), the interaction between prediction error and the two factors from the normative model, CPP ($\Omega_t$) and RU ($\tau_t$), as well as the interaction between prediction error and whether the outcome was rewarded or not[2]. The form of the regression model can be written as

$$\text{Update}_t = \beta_0 + \beta_1 \delta_t + \beta_2 \delta_t \Omega_t + \beta_3 \delta_t (1 - \Omega_t)\tau_t + \beta_4 \delta_t \text{Reward}_t + \beta_5 \text{Edge}_t + \varepsilon, \tag{6}$$

where Edge is regressor of no interest that captures the tendency to avoid updating toward the edges of the screen $((150 - B_{t+1})|150 - B_{t+1}|)$. If subjects used a fixed learning rate (Eq. (2) alone), $\beta_2$ and $\beta_3$ will be zero and $\beta_1$ will reflect that fixed learning rate. In contrast, if subjects behave exactly in accordance with the normative model (Eq. (3)), $\beta_2$ and $\beta_3$ will be one, and $\beta_1$ will be zero. Thus, we constructed the regression model so that the weights on $\beta_2$ and $\beta_3$ reflect the degree to which the two normative factors, CPP and RU, drive dynamic learning rates.

This regression model was fitted separately to each participant's data to estimate the influence of each factor on each participant's behavior. We used the residuals of this regression to examine the relationship between subgraph expression and residual updating. To examine the relationship between individual differences in normative learning and functional network dynamics, we used the sum of the regression coefficients on the CPP term ($\beta_2$) and the RU term ($\beta_3$) as an index of normative learning.

**MRI data acquisition and preprocessing**. MRI data were collected on a 3 T Siemens Trio with a 32-channel head coil. Functional data were acquired using gradient-echo echoplanar imaging (EPI) (3 mm isotropic voxels, 64 × 64 matrix, 42 axial slices tilted 30° from the AC–PC plane, TE = 25 ms, flip angle = 75°, TR = 2500 ms). There were 4 runs with 226 images per run. T1-weighted MPRAGE structural images (0.9375 × 0.9375 × 1 mm voxels, 192 × 256 matrix, 160 axial slices, TI = 1100 ms, TE = 3.11 ms, flip angle = 15°, TR = 1630 ms) and matched fieldmap images (TE = 2.69 and 5.27 ms, flip angle = 60°, TR = 1000 ms) were also collected. Data were preprocessed with FSL[37,38] and AFNI[39,40]. Functional data were corrected for slice timing (AFNI's 3dTshift) and head motion (FSL's MCFLIRT), attenuated for outliers (AFNI's 3dDespike), undistorted and warped to MNI space (FSL's FLIRT and FNIRT), smoothed with 6 mm FWHM Gaussian kernel (FSL's fslmaths) and intensity scaled by the grand-mean value per run. Structural images were segmented into gray matter, white matter (WM) and cerebrospinal fluid (CSF) (FSL's FAST)[41].

**Constructing time-varying functional networks**. For each run and each participant, BOLD time series were obtained from each of 264 ROIs (diameter = 9 mm) based on the previously defined parcellation[27]. ROIs that did not have valid BOLD time series for all runs and all participants were removed, resulting in N = 247 ROIs. We visualized these ROIs on the brain using the BrainNet Viewer (https://www.nitrc.org/projects/bnv)[42]. For each BOLD time series, a band-pass filter was applied with a cutoff of 0.01–0.08 Hz. This low-frequency band has been shown to reflect neuronal activation and neural synchronization[43–45]. To remove the influence of head motion, a confound regression was implemented to regress out nuisance factors from each BOLD time series. This confound regression included 24 motion parameters (three translation and three rotation motion parameters and

their expansion $([R_t R_t^2 R_{t-1} R_{t-1}^2])$[46], as well as average signals from WM and CSF[47].

In order to construct dynamic functional networks, we defined sliding time windows and calculated Pearson correlation coefficients between ROI time series in each sliding time window. We assigned these coefficients to the first TR in the time windows. To ensure magnetization equilibrium, the first 6 volumes of each run were removed from the analysis. For the rest of the volumes in each run, a sliding window was defined with a 10-TR (25 s) length and 80% overlap across windows. Each run had 106 sliding time windows, leading to $T = 424$ sliding time windows for each participant. Each participant's data thus formed a matrix of dynamic functional networks with dimensions $N \times N \times T$. Then, we took each participant's $N \times N$ matrix and unfurled the upper triangle into an $\frac{N(N-1)}{2}$ vector. By concatenating vectors across all time windows ($T$), we obtained an $\frac{N(N-1)}{2} \times T$ matrix. Furthermore, we concatenated matrices from $S = 32$ participants to form a $\frac{N(N-1)}{2} \times (T \times S)$ matrix. To ensure that our approach did not give undue preference to either positively or negatively weighted functional edges, we separated this matrix into two thresholded matrices: one composed of positively weighted edges, and one composed of negatively weighted edges. That is, in the matrix of positive functional correlations between ROI time series, the original negative correlations between ROI time series were set to 0; in the matrix of negative functional correlations between ROI time series, all values were multiplied by −1, and the original positive functional correlations between ROI time series were set to 0. After concatenating the matrix composed of positively weighted edges and the matrix of negatively weighted edges, we had a final $\frac{N(N-1)}{2} \times (T \times S \times 2)$ matrix $\mathbf{A}$.

**Clustering functional networks into subgraphs.** We applied an unsupervised machine learning algorithm called NMF[23] on $\mathbf{A}$ to identify subgraphs $\mathbf{W}$ and the time-dependent expressions of subgraphs $\mathbf{H}$. The matrix factorization problem $\mathbf{A} \approx \mathbf{WH} \ s.t. \mathbf{W} \geq 0, \mathbf{H} \geq 0$ was solved by optimization of the cost function

$$\min_{\mathbf{W},\mathbf{H}} \frac{1}{2}||\mathbf{A} - \mathbf{WH}||_F^2 + \alpha||\mathbf{W}||_F^2 + \beta \sum_{t=1}^{TS} ||\mathbf{H}(:,t)||_1^2, \quad (7)$$

where $\mathbf{A}$ is the functional connectivity matrix, $\mathbf{W}$ is a matrix of subgraph connectivity with size $\frac{N(N-1)}{2} \times k$, and $\mathbf{H}$ is a matrix of time-dependent expression coefficients for subgraphs with size $k \times (T \times S \times 2)$. The parameter $k$ is the number of subgraphs, $\alpha$ is a regularization of the connectivity for subgraphs, and $\beta$ is a penalty that imposes sparsity on the temporal expression coefficients[48]. For fast and efficient factorization to solve this equation, we used an alternative non-negative least square with the block-pivoting method with 100 iterations[49]. The matrices $\mathbf{W}$ and $\mathbf{H}$ were initialized with randomized values from a uniform distribution between 0 and 1.

A random sampling procedure was used to find the optimal parameters $k$, $\alpha$, and $\beta$[50]. In this procedure, the NMF algorithm was re-run 1000 times with parameter $k$ drawn from $U(2, 15)$, parameter $\alpha$ drawn from $U(0.01, 1)$, and parameter $\beta$ drawn from $U(0.01, 1)$. The subgraph learning performance was evaluated through four-fold cross-validation. In each fold, twenty-four participants were used for training; eight participants were used for testing and calculating cross-validation error ($||\mathbf{A} - \mathbf{WH}||_F^2$). An optimal parameter set should minimize the cross-validation error. We chose an optimal parameter set ($k = 10$, $\alpha = 0.535$, $\beta = 0.230$) that ensured the cross-validation error in the bottom 25% of the distribution of cross-validation error from our random sampling scheme[25].

Since the result of NMF is non-deterministic, we implemented consensus clustering to obtain reliable subgraphs[51]. In this procedure, we (i) used the optimal parameters and ran the NMF 100 times on $\mathbf{A}$, (ii) concatenated subgraph matrix $\mathbf{W}$ across 100 runs into an aggregate matrix with dimensions $\frac{N(N-1)}{2} \times (k \times 100)$, (iii) applied NMF to this aggregate matrix to obtain a final set of subgraphs $\mathbf{W}_{\text{consensus}}$ and expression coefficients $\mathbf{H}_{\text{consensus}}$.

**Properties of subgraphs.** Applying NMF yielded a set of subgraphs, or patterns of functional connectivity ($\mathbf{W}$), and the expression of these subgraphs over time ($\mathbf{H}$). To understand the subgraphs, we first rearranged $\mathbf{W}$ into $k$ different $N \times N$ subgraphs. To understand the roles of cognitive systems in each subgraph, we mapped each ROI to 13 putative cognitive systems from the previously defined parcellation: uncertain, sensory, cingulo-opercular task control, auditory, default mode, memory retrieval, visual, fronto-parietal task control, salience, subcortical, dorsal attention, ventral attention, and cerebellar[24,27]. This yielded a $13 \times 13$ representation of each subgraph. To show which within-system and between-system edges in this representation were strongest, we applied a permutation test. We permuted the system label for ROIs and formed a matrix with system-by-system edges. This process was repeated 10,000 times to determine which strength of system-by-system edges was above the 95% confidence interval threshold after correction for multiple comparisons.

To characterize the connectivity pattern of each subgraph, we ordered them according to the relative strength of within-system edges versus between-system edges. For each subgraph, we calculated the average strength of within-system edges (edges that link two ROIs that both belong to the same system), and the average strength of between-system edges (edges that link an ROI in one system to

an ROI in another system). Then, we subtracted the average strength of between-system edges ($E_B$) from the average strength of within-system edges ($E_W$) and divided this difference by the sum of them ($\frac{E_W - E_B}{E_W + E_B}$). We estimated the 95% confidence interval of these measures (average relative strength, average within-system strength or average between-system strength) by implementing bootstrapping 10,000 times.

Next, we investigated the relationship between these connectivity patterns and the temporal expression of each subgraph. As the matrix of functional connectivity was divided in two, with the first half reflecting positive connectivity and the second half reflecting negative connectivity, the temporal expression matrix also had two halves, with the first reflecting positive expression over time and the second reflecting negative expression over time. As there was a strong negative correlation between positive and negative expression, we did all of our analyses on the relative expression (positive expression minus negative expression) of each subgraph[26]. Across subgraphs, we calculated Pearson correlation coefficients between the average relative expression and the average within-system strength, average between-system strength, and average relative strength of each subgraph. To determine the significance of the correlation coefficients, we implemented 10,000 permutations of the subgraph labels to form the null distribution of correlation coefficients.

**Modulation of subgraph expression by task factors.** We investigated how fluctuations in the trial-by-trial relative expression of each subgraph were related to four trial-by-trial task factors: CPP, RU, reward, and residual updating. CPP and RU were estimated based on the normative learning model[1–3]. Residual updating was derived as the residual of the behavioral regression model described above. We examined the effect of these four trial-by-trial task factors together, including all four in a regression model predicting trial-by-trial relative expression. Since NMF yielded values of temporal expression every 2 TRs (5 s), we applied a linear interpolation on the temporal expression values to obtain an expression value aligned with outcome onset on each trial. Regression models were implemented for each participant separately. Regression coefficients were then tested at the group level using two-tailed $t$ tests.

**Association of individual learning with subgraph expression.** Next, we examined the relationship between subgraph expression and individual differences in the extent to which belief updating followed normative principles. As an index of normative learning for each individual, we used the sum of the regression coefficients on the CPP term ($\beta_2$) and the RU term ($\beta_3$) in the behavior model[2]. This normative learning index reflected the extent to which a participant's trial-by-trial updates were influenced by the two normative factors CPP and RU. We examined the relationship between this index and two aspects of subgraph expression. First, across subjects, we calculated the Pearson correlation coefficient between normative learning and the dynamic modulation of relative expression by normative factors for each subgraph. This dynamic modulation was indexed as the sum of the regression coefficients for CPP and RU from the regression model predicting trial-by-trial relative expression. That is, dynamic modulation reflected how normative factors were associated with the change in relative expression of the subgraph. Second, across subjects, we calculated the Pearson correlation coefficient between normative learning and the average relative expression of each subgraph. To determine the significance of these correlation coefficients, we permuted the participant labels 10,000 times to form the null distribution.

**Contribution of specific edges.** We evaluated the contributions of different types of edges to the task effects (influence of CPP, RU, reward and residual updating on subgraph expression across time) and individual differences effects (relationship between normative learning and subgraph expression across subjects). We mainly focused on the contribution of within-system edges and between-system edges. For this analysis, we implemented three types of comparison: Within versus All, Between versus All, and Between versus Within. For Within versus All, we kept within-system edges only and re-estimated task and individual differences effects; then, we compared these effects with the effects estimated using all edges. This comparison showed the change of effects after between-system edges were removed, and thus, this comparison revealed the contribution of between-system edges. For Between versus All, we kept between-system edges only and re-estimated task and individual differences effects. We then compared these effects with the effects estimated using all edges. In this comparison, within-system edges were removed and thus, we examined the contribution of within-system edges. Last, the comparison of Between versus Within is a direct comparison between effects estimated with between-system edges only and effects estimated with within-system edges only. Thus, this comparison examined the different contributions of between-system and within-system edges.

Specifically, for task effects, we examined the change of coefficients in the regression model that investigated the influence of four task factors—CPP, RU, reward and residual updating—on subgraph relative expression. The change was calculated for each participant separately, and the significance of change was then tested at the group level using two-tailed $t$ tests. For individual differences effects, we examined the change of correlation coefficients for two types of relationship: the relationship between individual normative learning and dynamic modulation of

subgraph relative expression and the relationship between individual normative learning and average subgraph relative expression. To determine the significance of the change of correlation coefficients, we permuted the labels of participants for individual normative learning 10,000 times to form the null distribution of the change of correlation coefficients.

We also investigated the contribution of different functional systems and the contribution of different system-by-system edges. For the contribution of different functional systems, we compared the effects after removing edges of one functional system with the effects estimated with all edges. For the contribution of different system-by-system edges, we compared the effects after removing one system-by-system edge with the effects estimated with all edges. Statistical testing was conducted with the same procedures described in the previous paragraph.

**Relationship between regional activity and connectivity**. To investigate the relationship between dynamic functional connectivity and univariate activation, we fit a mass univariate GLM. In this GLM, the regressors were the outcome onset and four modulators of outcome onset: CPP, RU, reward and residual updating. These regressors were convolved with a gamma hemodynamic response function (HRF) as well as the temporal derivative of this function. Six motion parameters were also included as regressors.

To examine what aspects of our functional connectivity results could be accounted for by functional coactivation, we used the regression coefficients from the GLM above (including both the main HRF and its temporal derivative for each regressor) to create a predicted BOLD time series. We then repeated the same sequence of analyses described above on this predicted BOLD time series. This predicted BOLD time series captured all fluctuations in activity in that ROI that could be accounted for by the linear effects of CPP, RU, reward, and residual updating. However, this predicted BOLD time series lacked any statistical dependencies between regions that were present in the actual BOLD time series that could not be explained by task-driven changes in univariate activation. Thus, any functional connectivity results we observed with this predicted BOLD time series could be fully accounted for by task-driven changes in univariate activation.

**Reporting summary**. Further information on research design is available in the Nature Research Reporting Summary linked to this article.

## Data availability
The data for the current study are available from the corresponding author upon request. The source data underlying Figs. 4c–e, 5, and 6a, b and Supplementary Figs. 3, 5–7, 10–12, 14–16, 18–20 are provided as a Source Data file.

## Code availability
Code is available at https://github.com/changhaokao/nmf_network_learning.

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

## Acknowledgements

This work was supported by grants from National Institute of Mental Health (R01-MH098899 to J.I.G. and J.W.K.) and National Science Foundation (1533623 to J.I.G. and J.W.K.). The funders had no role in study design, data collection and analysis, decision to publish or preparation of the paper.

## Author contributions

C.-H.K. and J.W.K. designed the study. M.R.N., J.T.M., J.I.G., and J.W.K. designed the task. M.R.N. and J.T.M. collected the data. A.N.K. established the software for NMF. A.N.K and D.S.B. provided suggestions in network analyses. C.-H.K. implemented all the analyses and visualization and drafted the paper. C.-H.K., A.N.K., D.S.B., M.R.N., J.T.M., J.I.G., and J.W.K. interpreted the results and revised the paper.

## Competing interests

The authors declare no competing interests.
