## [Peer Review File · Nature Communications]

Reviewers' Comments:

Reviewer #1:

Remarks to the Author:

Kao et al. report a reanalysis of a previous fMRI study which investigates the reconfiguration of functional brain networks during learning in a dynamic environment. Participants had performed a learning task in which they had to update their beliefs depending on their uncertainty and the likelihood that a major change in the environment occurred. Their learning behavior was modeled using a normative learning model in which change-point probability (CPP) and relative uncertainty (RU) drive the learning rate. Task-irrelevant rewards also influenced learning rate although they were not captured by the normative model.

In the present analysis the authors identified patterns of whole-brain functional connectivity that changed dynamically over time and were more strongly expressed when surprise and uncertainty (CPP, RU) were high and enhanced in individuals whose belief updating was more strongly driven by these factors. This pattern of functional connectivity involved stronger functional integration of the frontoparietal network and other functional systems.

The manuscript presents interesting and novel results and demonstrates a sophisticated way of analyzing whole-brain functional connectivity. The methodological approach is rigorous and thorough and the authors provide a large set of additional analyses to demonstrate the robustness of the findings. In my view, the impact of this manuscript could be enhanced by extending the discussion of the findings and linking them more to the brain activity findings (McGuire et al., 2014) as well as an RSA-based analysis of the same data suggesting "network resets" (state changes) in some brain areas related to RU and CPP (Nassar et al., 2018). For readers less familiar with the topic it might be difficult to appreciate the significance of the findings and their relationship to these previous reports. Briefly, can we, based on the current network findings, come up with hypotheses how certain functional networks interact during learning and associated state changes?

I was surprised that the subgraph that responded most to CPP and RU seemed to have a rather weak connection/overlap with the cinguloopercular network, given that dorsal anterior cingulate, insular and (medial) parietal cortices seemed to form the core network in McGuire et al. (2014) that represented learning rate and connected to other brain regions representing RU, CPP and reward, respectively, when these factors influenced learning rate most.

The authors compare their findings with a second univariate GLM analysis related to CPP and RU. While some of the findings using NMF on the "raw" fMRI time series were also captured by this univariate analysis, others were not. What does this mean? The discussion suggests that this difference may support the notion that the revealed network dynamics indicate an increase in neural communication. I do not understand this argument and would like to ask the authors to elaborate more on this.

It would also be nice, if the authors could present some justification for the use of NMF. There are many unsupervised machine learning algorithms around that can provide blind source separation. Does NMF have advantages over other algorithms (e.g., ICA) and, if so, which?

Would it make sense to also investigate and interpret other subgraphs whose temporal expression appears to be partly influenced by some of the relevant factors (CPP, RU), as shown in Suppl. Fig. 3?

Reviewer #2:

Remarks to the Author:

In this paper the authors use a decision task, fMRI, and non-negative matrix factorization (NMF) to examine how networks of brain activity (defined by NMF) are modulated through time by computational variables defined by a normative model of the decision task.

By fitting the NMF decomposition (regularized) of the vectorized and sign-separated correlation matrix of (fMRI) activity from 247 (usable) ROIs, the authors obtained 10 subgraphs (features of the decomposition). For summarizing properties, these subgraphs were then converted into subgraphs based on 13 "putative cognitive systems". The main results concerned the relationship between quantities used to model behavior in the task, and the time series created as a result of the NMF.

First, the time series of one subgraph, subgraph 4, was more related to task-related quantities than the other subgraphs (the task quantities were CPP, change point probability or "belief surprise", and RU, relative uncertainty or "belief uncertainty", reward, and the residual of a behavioral regression of belief changes on the prediction error of the normative model, as well as interaction terms and reward). The first result was normative learning (sum of regression coefficients of CPP and RU in the behavioral model) and was positively correlated with "dynamic modulation", the sum of the regression coefficients of the regression of the subgraph expression on CPP and RU, as well as the average expression of subgraph 4.

The next result concerned the changes of the various quantities relating CPP and RU with respect to the omission of between or within edges in the subgraph – this was repeated for the "within-subject effects" (regression coefficients of CPP, RU, etc.) and the "between-subject effects" (correlation of behavioral and expression regression coefficients of RU and CPP as well as correlation of average expression and the sum of the coefficients of RU and CPP in the behavioral regression). The final main result concerns the relationship between the univariate neural results and the results of this NMF analysis. A measure of edge strength for subgraph four ROIs correlated positively with z-scores from the regressors for CPP and RU in the univariate model.

General.

The study of the dynamics of neural networks (real ones, not artificial ones) is certainly important. The use of a compelling task, computational modelling, and a method of decomposing the dynamics of the networks (NMF) – and then relating the resulting time series to behavioral quantities is a reasonable strategy. However, the exposition is so disjointed that it is difficult to understand what has actually been done. The general feeling I get is that there is so much going on and that the authors are so familiar with the material that they have lost sight of having to communicate with an audience that might be less familiar with the material. This is exacerbated by the lead to lean on the original paper (the univariate results and normative model).

Major Specific Points.

1. The terms "within-subject effects" and "between subject effects" are not used until the methods. They do not appear in the results of figure legends. This can lead to confusion.
2. The text refers to the behavioral regression coefficients of CPP and RU, yet the Methods has a regression with no such terms:
3. The text, figure legends (Figure 5), and Methods refer to re-estimating the within-subject and between-subject relationships by removing within-system and between-system edges. This move

needs to be explained in more detail.

4. Please clarify upfront the definition of subgraph expression as it relates to the positive and negative parts (is this definition used in all instances of the use of the term?).

5. I felt that the narrative and the scope of the main claims got lost in this version of the manuscript and I recommend seriously considering a careful re-reading and re-writing to improve clarity of the claims.

Reviewer #3:

Remarks to the Author:

I enjoyed reading this manuscript reporting an analysis of how patterns of connectivity between brain regions change in tandem with two factors (sometimes referred to as surprise and uncertainty) that ought, normatively, to determine the speed of learning. The authors used an interesting way to identify patterns of interregional activity correlation that will be novel to many but which is explained very clearly and succinctly. I have suggested a few other pieces of information that the authors might provide to help readers understand their approach (point 1). Variation in activity in this inter-regional network (or subgraph) is then linked to variation in the factors determining learning across the task and variation across subjects. Final analyses confirm that these effects are not just attributable to interactions within the well-known "resting state" networks that have been defined previously or to univariate brain activity changes. Although my comments are prolix they are minor.

1 P4 The description of the method used to identify patterns of interregional activity correlation is explained in the last paragraph on page 4. Figures 2 and 3 are very helpful and the method is admirably clearly but also succinctly explained up to the sentence "The full data matrix was divided into two halves, with one half containing the positive correlation coefficients (zero if the coefficient was negative) and one half containing the absolute values of negative correlation coefficients (zero if the coefficient was positive)²⁴" but the next sentence "We applied NMF to this matrix to identify functional subgraphs and their expression over time", which is critical and which recurs figure legends could be clarified further at least somewhere near this point in the main text. The only other information about what constitutes a subgraph is in the Introduction: "patterns of functional connectivity across the entire brain". A few more words of explanation either at the bottom of page 4 in the Results or in the Introduction would help. Presumably a subgraph consists of nodes with timeseries of activity that exceed some threshold of correlation or other measure of similarity? Are they based only on positive correlation values? Can a given subgraph be identified on the basis of both positive correlations and the magnitude of negative correlations? Presumably there is some method or threshold for determining when subgraph is significantly separate and different from another and therefore a meaningful entity? It seems that they are distinct entities to the functional systems that are summarized in figure 2a on the basis of the Power study. However, a clear statement to this effect outside the Methods is likely to help many readers.

2 Figure 4b illustrates the subgraphs – the correlated patterns of activity across regions – that were identified in a series of lateral views. Given the importance of subgraph 4 – the subgraph that seems most closely related to the variation in updating – it would be useful to also have a medial view illustration of subgraph 4. It might be helpful to clarify where some of the activity is found. For example, the subgraph is described as being frontoparietal which for some people will seem a very broad term given the limited number of lobes that the brain consists of. For others it may conjure up an image of the posterior lateral prefrontal cortex and intraparietal region. However, such an impression would not be quite correct; it looks as if it might be, within parietal cortex, predominantly

the inferior parietal lobule that contributes to subgraph 4 and it looks as if the cingulate and frontal opercular regions are as important as lateral prefrontal cortex.

3 End of paragraph 1. "On the neural level, many studies have shown that such uncertainty and surprise in a dynamic environment was represented as univariate and multivariate activity in medial and lateral fronto-parietal networks^{1, 4, 5, 6}." Most of the studies cited here involve univariate fMRI analyses. Perhaps the recent multivariate analysis by Meder and colleagues (Nature Communications, 2017) might be cited here? The study seems relevant because, while not detracting from the novelty of the current manuscript, it also reports changes in correlation between multivariate patterns in four areas that appear to be have some overlap with those identified with subgraph 4 (although this could be clarified – see point 2).

Typos

"participants' predictions were influence by..."

Reviewers' comments:

Reviewer #1 (Remarks to the Author):

Kao et al. report a reanalysis of a previous fMRI study which investigates the reconfiguration of functional brain networks during learning in a dynamic environment. Participants had performed a learning task in which they had to update their beliefs depending on their uncertainty and the likelihood that a major change in the environment occurred. Their learning behavior was modeled using a normative learning model in which change-point probability (CPP) and relative uncertainty (RU) drive the learning rate. Task-irrelevant rewards also influenced learning rate although they were not captured by the normative model.

In the present analysis the authors identified patterns of whole-brain functional connectivity that changed dynamically over time and were more strongly expressed when surprise and uncertainty (CPP, RU) were high and enhanced in individuals whose belief updating was more strongly driven by these factors. This pattern of functional connectivity involved stronger functional integration of the frontoparietal network and other functional systems.

The manuscript presents interesting and novel results and demonstrates a sophisticated way of analyzing whole-brain functional connectivity. The methodological approach is

rigorous and thorough and the authors provide a large set of additional analyses to demonstrate the robustness of the findings.

Comment 1:

In my view, the impact of this manuscript could be enhanced by extending the discussion of the findings and linking them more to the brain activity findings (McGuire et al., 2014) as well as an RSA-based analysis of the same data suggesting "network resets" (state changes) in some brain areas related to RU and CPP (Nassar et al., 2018). For readers less familiar with the topic it might be difficult to appreciate the significance of the findings and their relationship to these previous reports. Briefly, can we, based on the current network findings, come up with hypotheses how certain functional networks interact during learning and associated state changes?

Response:

We thank the reviewer for their positive feedback and for this suggestion. We have expanded the section of the discussion that considers the relationship between the current analyses and our previous reports (McGuire et al., 2014; Nassar et al., 2019). In the discussion (see paragraph 1-2 of "Relationship to previous results: neural activation during learning in dynamic environments" on p.10-11), we provide more detail on how we think these results are linked.

(paragraph 1-2 of "Relationship to previous results: neural activation during learning in dynamic environments" on p.10-11")

The brain-wide pattern of functional connectivity that we identified as connected to belief updating included interactions between multiple functional systems, most prominently the fronto-parietal task control, memory-retrieval, salience, and dorsal-attention systems (Fig. 3b & Fig. 4d). These systems include multiple regions in the anterior insula, dorsolateral and dorsomedial frontal cortex, and lateral and medial parietal cortex (Fig. 4e & Fig. 6c). These regions showed a large degree of overlap with areas that we have previously shown to have increased univariate activation in response to both surprise and uncertainty (in this same dataset; Figure 6)². The current results show that the same regions also exhibit increased functional connectivity in response to surprise and uncertainty. Our previous report also identified a subset of these regions, including parts of the dorsomedial frontal cortex, anterior insula, inferior frontal cortex, posterior cingulate cortex, and posterior parietal cortex, where activity was modulated by both normative (surprise and uncertainty) and non-normative (reward) factors. This smaller subset includes regions that participate in the fronto-parietal task-control, memory-retrieval, salience, and dorsal-attention systems, which is where we identified learning-related increases in functional connectivity in the current study.

Previously, we also reported regions whose univariate activity was modulated by either surprise or uncertainty alone. Surprise was associated selectively with activation in occipital cortex, and uncertainty was associated selectively with activation in anterior prefrontal and parietal cortex². We similarly have reported multivariate activation patterns that were associated selectively with either surprise or uncertainty alone⁷. In the current study, we identified a key pattern of functional connectivity that was

modulated by both surprise and uncertainty, but we did not identify any other pattern that was modulated reliably by either surprise or uncertainty alone. One possible explanation for this lack of a positive result was our need to use relatively long time windows (25 s, corresponding to 4-6 trials) in order to obtain reliable estimates of functional connectivity. These time windows likely included both the surprise elicited by change-points and the uncertainty that follows. Thus, functional connectivity related to surprise and uncertainty may have been difficult to dissociate temporally in our task and analysis design. Using a task that can temporally separate the tracking of surprise and uncertainty²⁸ might enable the identification of distinct patterns of functional connectivity for each factor.

Comment 2:

I was surprised that the subgraph that responded most to CPP and RU seemed to have a rather weak connection/overlap with the cinguloopercular network, given that dorsal anterior cingulate, insular and (medial) parietal cortices seemed to form the core network in McGuire et al. (2014) that represented learning rate and connected to other brain regions representing RU, CPP and reward, respectively, when these factors influenced learning rate most.

Response:

We showed the functional systems with strong connectivity strength in Fig. 4e and the regions with strong connectivity strength in Fig. 6c. In the discussion section (see paragraph 1 of “Relationship to previous results: neural activation during learning in dynamic environments” on p.10-11, quoted in response to comment 1 above), we also describe the extensive spatial overlap between the regions in the learning-related subgraph and the regions with strong activation for CPP and RU in McGuire et al. (2014). Based on the parcellation in Power et al. (2011), the overlapping regions corresponded to multiple functional systems: insula (salience), dorsomedial frontal cortex (cingulo-opercular, fronto-parietal, and salience), dorsolateral prefrontal cortex (fronto-parietal), posterior parietal cortex (fronto-parietal, salience, and dorsal attention) and occipital cortex (visual). Thus, we identified similar regions shown in McGuire et al. (2014) and these regions were located in multiple functional systems, including the cingulo-opercular system as well as others. In addition, the critical features of the learning-related subgraph (Fig. 3b) required the interactions between the fronto-parietal system and other systems (e.g., cingulo-opercular system).

Comment 3:

The authors compare their findings with a second univariate GLM analysis related to CPP and RU. While some of the findings using NMF on the "raw" fMRI time series were also captured by this univariate analysis, others were not. What does this mean? The discussion suggests that this difference may support the notion that the revealed network dynamics indicate an increase in neural communication. I do not understand this argument and would like to ask the authors to elaborate more on this.

Response:

In the discussion (see paragraph 4 of “Relationship to previous results: neural activation during learning in dynamic environments” on p.11-12), we now clarify and elaborate on this argument. The increased correlation between activity in different brain regions, which is reflected increased subgraph expression, could in principle be driven by two different types of “functional connectivity”: co-activation induced by task factors or correlation between regional activities that cannot be accounted for by task-driven activations. To evaluate the influence of task-driven co-activation, we repeated our NMF analysis using the predicted fMRI time series from a univariate GLM analysis of task-related factors. The correlations between regions in this predicted time series are entirely explained by task-driven co-activation, so if we replicate all of our findings with the raw fMRI time series in this predicted time series, then we would know that our findings are driven entirely by task-driven co-activation. However, while we replicate some of our findings with the raw fMRI time series in the predicted time series, we do not replicate all of them. Instead, critical effects of task factors (e.g., the modulation of subgraph expression by surprise and residual updating) and individual differences (e.g., the relationship between individual differences in normative learning and the dynamic modulation of subgraph expression by normative factors) are only found in the analyses of the raw time series. Thus, task-driven co-activation cannot explain all of our findings. Instead, this analysis suggests that some of our effects are driven by endogenous connectivity, rather than task-driven co-activation. Therefore, we conclude that the network dynamics we identified reflect not only task-driven co-activation but also neural communication between brain regions.

(paragraph 4 of “Relationship to previous results: neural activation during learning in dynamic environments” on p.11-12)

Functional connectivity captures many different kinds of statistical dependencies between brain regions, including those that result from task-driven co-activation. The strong association between neural activation and functional connectivity during periods of surprise and uncertainty in our results (Fig. 6), as well as previous studies in other domains^{13, 15, 17, 19, 21, 22}, raises the possibility that the increases in functional connectivity between brain regions might have arisen because these regions became more tightly synchronized to external task events, without necessarily any increase in communication between them. To refute this possibility, we repeated our analyses on the predicted BOLD time series from univariate GLMs. These predicted time series, which contain only task-driven statistical dependencies between brain regions, could not recapitulate all of the effects that we observed in our actual BOLD time series. Specifically, we found modulations by task (e.g., the modulation of subgraph expression by surprise and residual updating) and individual differences (e.g., the relationship between individual differences in normative learning and the dynamic modulation of subgraph expression by normative factors) that were apparent only in the full, original functional connectivity matrices. Thus, these effects appear to include neural communications that do not simply reflect task-driven co-activation. Even though the changes in functional connectivity that we describe may reflect a mixture of task-driven and endogenous dynamics, the network analysis provides an important higher-level, reduced-dimensionality description of these changes.

Comment 4:

It would also be nice, if the authors could present some justification for the use of NMF. There are many unsupervised machine learning algorithms around that can provide blind source separation. Does NMF have advantages over other algorithms (e.g., ICA) and, if so, which?

Response:

We added to the introduction (see paragraph 3 of introduction on p.3-4) a more extensive rationale for the use of NMF, as opposed to other unsupervised learning algorithms that can provide blind source separation. We argue that NMF has two advantages over other approaches such as PCA or ICA. First, NMF yields a parts-based representation of the network, in which the individual components are strictly additive – a constraint that is not present in PCA and ICA. This important feature enables interpretation of the resulting subgraph and time-dependent expression coefficients on the basis of their positive distance from zero. Second, NMF does not enforce an orthogonality or independence constraint and, therefore, allows subgraphs to overlap in their structure. This property may more effectively model distinct subgraphs that may be jointly related via weak connections and better account for the flexibility of neural systems, in which one region can be simultaneously involved in multiple systems or cognitive functions.

(paragraph 3 of Introduction on p.3-4)

In the current study, we aimed to identify such frequent reconfigurations in functional connectivity during adaptive belief updating. A key to our approach was the use of an unsupervised machine-learning technique known as non-negative matrix factorization (NMF)²³. NMF decomposes the whole-brain network into subgraphs, which describe patterns of functional connectivity across the entire brain, and the time-dependent magnitude with which these subgraphs are expressed. Briefly, a subgraph is a weighted pattern of functional interactions that statistically recurs as the brain network evolves over time. We chose NMF because it provides two key advantages over other approaches to matrix factorization, such as principal components analysis (PCA) or independent components analysis (ICA)^{24, 25}. First, NMF yields a parts-based representation of the network, in which the individual components are strictly additive – a constraint that is not present in PCA and ICA. This important feature enables interpretation of the resulting subgraph and time-dependent expression coefficients on the basis of their positive distance from zero. Second, NMF does not enforce an orthogonality or independence constraint and, therefore, allows subgraphs to overlap in their structure. This property may more effectively model distinct subgraphs that may be jointly related via weak connections and better account for the flexibility of neural systems, such that one connection between regions can be involved in multiple systems or cognitive functions. Recently, NMF has been used to identify network dynamics during rest and task states^{25, 26} and to determine how these dynamics vary across development²⁴. Here we extend the use of this technique to examine changes in network dynamics linked to task variables and individual differences.

Comment 5:

Would it make sense to also investigate and interpret other subgraphs whose temporal expression appears to be partly influenced by some of the relevant factors (CPP, RU), as shown in Suppl. Fig. 3?

Response:

We have sought to balance a complete reporting of our results in the supplement with a focused discussion and interpretation of only the most reliable results in the manuscript. There are several reasons why we have decided to keep the focus on one subgraph (subgraph 4) in the results (see paragraph 2 of “Normative factors that drive belief updating modulated the temporal expression of a learning-related subgraph” on p.7), even though other subgraphs (e.g., subgraphs 1, 3 and 7) also showed effects of CPP or RU. First, among the ten subgraphs, CPP and RU explained the most variance in the temporal expression of subgraph 4 (Supplementary Fig. 2g). Second, the effects of CPP and RU in subgraph 4 were strong enough to survive correction for multiple comparisons across ten subgraphs. Third, the effects of CPP and RU on subgraph 4 were robust across analyses with different sizes of time window (see the section of “Results are robust to the time window used to identify functional connectivity“). Although we focus the discussion and interpretation in the manuscript on subgraph 4, we also include the results of all analyses for all ten subgraphs in the supplementary figures, for the benefit of interested readers.

(paragraph 2 of “Normative factors that drive belief updating modulated the temporal expression of a learning-related subgraph” on p.7)

Although CPP or RU also modulated the relative expression of some other subgraphs (e.g., subgraph 1, 3 and 7; Supplementary Fig. 3), below we focus on subgraph 4 for several reasons. First, as stated above, the four factors we investigated (CPP, RU, reward and residual updating) explained more variance in the time-dependent relative expression of subgraph 4 than that of any other subgraph. Second, the effects of CPP and RU on subgraph 4 were strong enough to survive correction for multiple comparisons across ten subgraphs, whereas the effects of CPP and RU on other subgraphs were not. Third, as discussed below (see the section of “Results are robust to the time window used to identify functional connectivity“), the effects of CPP and RU on subgraph 4 were also robust across analyses that used differently sized time windows, whereas the effects of CPP and RU on other subgraphs were not.

Reviewer #2 (Remarks to the Author):

In this paper the authors use a decision task, fMRI, and non-negative matrix factorization (NMF) to examine how networks of brain activity (defined by NMF) are modulated through time by computational variables defined by a normative model of the decision task.

By fitting the NMF decomposition (regularized) of the vectorized and sign-separated correlation matrix of (fMRI) activity from 247 (usable) ROIs, the authors obtained 10 subgraphs (features of the decomposition). For summarizing properties, these

subgraphs were then converted into subgraphs based on 13 “putative cognitive systems”. The main results concerned the relationship between quantities used to model behavior in the task, and the time series created as a result of the NMF.

First, the time series of one subgraph, subgraph 4, was more related to task-related quantities than the other subgraphs (the task quantities were CPP, change point probability or “belief surprise”, and RU, relative uncertainty or “belief uncertainty”, reward, and the residual of a behavioral regression of belief changes on the prediction error of the normative model, as well as interaction terms and reward). The first result was normative learning (sum of regression coefficients of CPP and RU in the behavioral model) and was positively correlated with “dynamic modulation”, the sum of the regression coefficients of the regression of the subgraph expression on CPP and RU, as well as the average expression of subgraph 4.

The next result concerned the changes of the various quantities relating CPP and RU with respect to the omission of between or within edges in the subgraph – this was repeated for the “within-subject effects” (regression coefficients of CPP, RU, etc.) and the “between-subject effects” (correlation of behavioral and expression regression coefficients of RU and CPP as well as correlation of average expression and the sum of the coefficients of RU and CPP in the behavioral regression). The final main result concerns the relationship between the univariate neural results and the results of this NMF analysis. A measure of edge strength for subgraph four ROIs correlated positively with z-scores from the regressors for CPP and RU in the univariate model.

General.

The study of the dynamics of neural networks (real ones, not artificial ones) is certainly important. The use of a compelling task, computational modelling, and a method of decomposing the dynamics of the networks (NMF) – and then relating the resulting time series to behavioral quantities is a reasonable strategy.

Comment 1:

However, the exposition is so disjointed that it is difficult to understand what has actually been done. The general feeling I get is that there is so much going on and that the authors are so familiar with the material that they have lost sight of having to communicate with an audience that might be less familiar with the material. This is exacerbated by the lead to lean on the original paper (the univariate results and normative model).

Response:

We thank the reviewer for this feedback. In the revised manuscript, we have made many changes to improve the exposition. We have also now included any necessary details from the original paper so that readers do not have to know the details of that paper to follow the current exposition. In particular, we have provided more details in the methods sections on the normative behavioral model, and in the results section on the previous univariate analyses (see paragraph 2-3 of “Belief updating is influenced by

normative factors related to uncertainty and surprise” in the Results section on p.4-5; “Behavior model” in the Methods section on p.14-16).

(Result of paragraph 2-3 “Belief updating is influenced by normative factors related to uncertainty and surprise” on p.4-5)

In our previous reports^{2, 7}, we described a theoretical model approximating the normative solution for this task. This theoretical model takes the form of a delta-rule and approximates the Bayesian ideal observer. Beliefs (B_{t+1}) are updated based on past beliefs (B_t) and prediction errors ($X_t - B_t$; the difference between the current outcome location and the predicted location), with the extent of updating controlled by a learning rate (α_t ; Fig. 1b). Learning rates vary from trial to trial and are determined by two factors: (i) change-point probability (CPP), which is the probability that a change-point has happened and represents a form of belief surprise; and (ii) relative uncertainty (RU), which is the reducible uncertainty regarding the current state relative to the irreducible uncertainty that results from environmental noise and represents a form of belief uncertainty (Fig. 1c). Learning rates are higher when either CPP or RU is higher: $\alpha_t = CPP + RU(1 - CPP)$.

We previously reported how participants’ predictions were influenced by both normative and non-normative factors and how these factors that influence learning are encoded in univariate and multivariate activity^{2, 7}. Participants updated their beliefs more when the value of CPP or RU was higher, consistent with the normative model. Participants also updated their beliefs more when the outcome was rewarded, however, which is not a feature of the normative model. CPP, RU and reward, as well as residual updating (belief updating not captured by CPP, RU or reward), were all encoded in univariate and multivariate brain activity in distinct regions^{2, 7}. In the current study, we built on these previous findings and investigated how these factors, as well as individual differences in how these factors influence belief updating, are related to the dynamics of whole-brain functional connectivity.

(Method of “Behavior model” on p.14-16)

We applied the same normative model described in our previous study². An approximation to the ideal observer solution to this task updates beliefs according to a delta learning rule (Fig. 1b):

$$\delta_t = X_t - B_t \quad (1)$$

$$B_{t+1} = B_t + \alpha_t \times \delta_t \quad (2)$$

where δ_t is the prediction error, which is the difference between the observed outcome (bag drop location, X_t) and the prediction (bucket location, B_t). Beliefs are updated in proportion to the prediction error, and this proportion is determined by α_t , the learning rate. The learning rate is adjusted adaptively on each trial according to two normative factors (Fig. 1c):

$$\alpha_t = \Omega_t + (1 - \Omega_t) \times \tau_t \quad (3)$$

where Ω_t is the change-point probability (CPP) and τ_t is the relative uncertainty (RU). CPP reflects the likelihood that a change-point has happened:

$$\Omega_t = \frac{U(X_t|0, 300)^H}{U(X_t|0, 300)^H + N(X_t|B_t, \sigma_t^2)^{(1-H)}} \quad (4)$$

where $U(X_t|0, 300)$ indicates the probability of X_t from a uniform distribution between 0 and 300, $N(X_t|B_t, \sigma_t^2)$ indicates the probability of X_t from a Gaussian distribution with mean of B_t and variance of σ_t^2 , σ_t^2 is the variance of predictive distribution of the bag location, and H is the average probability of change (0.1) across trials.

RU reflects the uncertainty about the current location of the helicopter relative to the amount of noise in the environment:

$$\tau_{t+1} = \frac{\Omega_t \sigma_N^2 + (1 - \Omega_t) \tau_t \sigma_N^2 + \Omega_t (1 - \Omega_t) (\delta_t (1 - \tau_t))^2}{\Omega_t \sigma_N^2 + (1 - \Omega_t) \tau_t \sigma_N^2 + \Omega_t (1 - \Omega_t) (\delta_t (1 - \tau_t))^2 + \sigma_N^2} \quad (5)$$

where σ_N^2 is the variance of outcome distribution used to generate the location of bag. There are three terms present in both the numerator and denominator. The first term is the variance of the helicopter distribution conditional on a change-point while the second term is the variance of the helicopter distribution conditional on no change-point. The third term reflects the variance due to the difference in mean between the two conditional distributions. The three terms together capture the uncertainty about the location of the helicopter.

Figure 1c shows an example of the dynamics of CPP and RU. CPP increases when there is an unexpectedly large prediction error. RU increases after CPP increases and decays slowly as more precise estimates of the helicopter location are possible.

As in our previous study, a regression model was applied to investigate how the factors in this normative model, as well as other aspects of the task, influenced participants' belief updates. We regressed trial-by-trial updates ($B_{t+1} - B_t$) against the prediction error (δ_t), the interaction between prediction error and the two factors from the normative model, CPP (Ω_t) and RU (τ_t), as well as the interaction between prediction error and whether the outcome was rewarded or not². Note that these interaction terms test whether learning rates vary as a function of CPP, RU or reward; if these coefficients are zero, updates would be a constant fraction of prediction errors (given by the β_1 term). The form of the regression model can be written as

$$Update_t = \beta_0 + \beta_1\delta_t + \beta_2\delta_t\Omega_t + \beta_3\delta_t\tau_t(1 - \Omega_t) + \beta_4\delta_tReward_t + \beta_5Edge_t + \varepsilon \quad (6)$$

where *Edge* is regressor of no interest that captures the tendency to avoid updating toward the edges of the screen ($(150 - B_{t+1})|150 - B_{t+1}|$). This regression model was fitted separately to each participant's data to estimate the influence of each factor on each participant's behavior. We used the residuals of this regression to examine the relationship between subgraph expression and residual updating. To examine the relationship between individual differences in normative learning and functional network dynamics, we used the sum of the regression coefficients on the CPP term (β_2) and the RU term (β_3) as an index of normative learning.

Comment 2:

The terms “within-subject effects” and “between-subject effects” are not used until the methods. They do not appear in the results of figure legends. This can lead to confusion.

Response:

To reduce confusion, we now replace “within-subject effects” and “between-subject effects” (terms that were not used until the methods) with “task effects” and “individual difference effects” (terms that were used beginning earlier, including in the results and figure legends, Fig. 4a-c) throughout the manuscript.

Comment 3:

The text refers to the behavioral regression coefficients of CPP and RU, yet the Methods has a regression with no such terms:

Response:

We now clarify by directly referring to the coefficients of the CPP term (β_2) and the RU term (β_3) in the behavioral regression (Eq. 6 on p.15).

Comment 4:

The text, figure legends (Figure 5), and Methods refer to re-estimating the within-subject and between-subject relationships by removing within-system and between-system edges. This move needs to be explained in more detail.

Response:

We added more details about this analysis and its motivation in the results (see paragraph 2 of “Contribution of specific edges within the learning-related subgraph to its relationship with task factors and individual differences” on p.8), figure legends (Fig. 5a), and methods (see paragraph 1-2 of “Contributions of different types of edges to task and individual differences effects” on p.19-20). The pattern of functional connectivity that we identify as being stronger during periods of surprise and uncertainty (“task effects”) and stronger in subjects that learn in a more normative manner (“individual difference effects”), involves increases in connectivity within functional systems and between functional systems. The motivation for this analysis is to determine whether

these effects are driven primarily by the within-system connectivity changes, the between-system changes, or both. Therefore, we re-estimated all of our effects (task and individual differences) using within-system edges only (Within), between-system edges only (Between), or all edges (All).

(paragraph 2 of “Contribution of specific edges within the learning-related subgraph to its relationship with task factors and individual differences” on p.8)

The task-related modulations of subgraph 4 involved primarily between-system, not within-system, functional connectivity. Specifically, we re-estimated the effects of CPP, RU, reward, and residual updating on the relative expression of subgraph 4 using only within-system edges (i.e., only the diagonal cells of the system-by-system matrix in Fig. 3b; “Within”) or only between-system edges (i.e., only the off-diagonal cells of the system-by-system matrix in Fig. 3b; “Between”). We compared these effects to our previous estimates using all edges (Fig. 5a; “All”). Removing the between-system edges (Within versus All) reduced the size of the estimated effects of CPP (mean±SEM=-0.155±0.042, $t_{37}=-3.73$, $p<0.001$), RU (-0.300±0.062, -4.82, $p<0.001$), and residual updating (-0.140±0.053, -2.63, $p=0.013$). In contrast, removing the within-system edges (Between versus All) led to no reliable changes in these effects (all $p>0.21$). Further, in a direct comparison of the reduced subgraphs with only within- or between-system edges, the effects estimated with between-system edges only were stronger for CPP (0.151±0.042, 3.63, $p<0.001$), RU (0.290±0.063, 4.63, $p<0.001$), and residual updating (0.139±0.048, 2.91, $p=0.007$).

(paragraph 1-2 of “Contributions of different types of edges to task and individual differences effects” on p.19-20)

We evaluated the contributions of different types of edges to the task effects (influence of CPP, RU, reward and residual updating on subgraph expression across time) and individual differences effects (relationship between normative learning and subgraph expression across subjects). We mainly focused on the contribution of within-system edges and between-system edges. For this analysis, we implemented three types of comparison: Within versus All, Between versus All, and Between versus Within. For Within versus All, we kept within-system edges only and re-estimated task and individual differences effects; then, we compared these effects with the effects estimated using all edges. This comparison showed the change of effects after between-system edges were removed, and thus, this comparison revealed the contribution of between-system edges. For Between versus All, we kept between-system edges only and re-estimated task and individual differences effects. We then compared these effects with the effects estimated using all edges. In this comparison, within-system edges were removed and thus, we examined the contribution of within-system edges. Last, the comparison of Between versus Within is a direct comparison between effects estimated with between-system edges only and effects estimated with within-system edges only. Thus, this comparison examined the different contributions of between-system and within-system edges.

Specifically, for task effects, we examined the change of coefficients in the regression model that investigated the influence of four task factors—CPP, RU, reward

and residual updating—on subgraph relative expression. The change was calculated for each participant separately, and the significance of change was then tested at the group level using two-tailed t -tests. For individual differences effects, we examined the change of correlation coefficients for two types of relationship: the relationship between individual normative learning and dynamic modulation of subgraph relative expression and the relationship between individual normative learning and average subgraph relative expression. To determine the significance of the change of correlation coefficients, we permuted the labels of participants for individual normative learning 10,000 times to form the null distribution of the change of correlation coefficients.

Comment 5:

Please clarify upfront the definition of subgraph expression as it relates to the positive and negative parts (is this definition used in all instances of the use of the term?).

Response:

We clarified that, in the analyses related to subgraph expression, we used relative expression by subtracting negative expression from positive expression at each time point. We provided more detail about how we calculate relative expression in the results section (see paragraph 5 of “NMF identified ten subgraphs, or patterns of whole-brain functional connectivity, that varied over time” on p.6), the methods section (see paragraph 3 of “Properties of subgraphs” on p.18), and the relevant figure legend (Fig. 4a).

(paragraph 5 of “NMF identified ten subgraphs, or patterns of whole-brain functional connectivity, that varied over time” on p.6),

The full data matrix on which we performed NMF was divided into two halves, with the first half corresponding to positive functional connectivity and the second half corresponding to negative functional connectivity. The expression matrix H was therefore also divided into two halves, with the first half corresponding to positive expression over time and the second half corresponding to negative expression over time. Positive and negative expression coefficients were highly negatively correlated with each other across time for all the subgraphs (all $r < -0.61$, all $p < 0.001$). Given this result, for the analyses of subgraph expression below, we constructed a measure of relative subgraph expression by subtracting the negative expression from the positive expression at each time point²⁶. Across subgraphs, the average relative expression across time was strongly correlated with the relative strength of within- versus between-system edges (Supplementary Fig. 2d-f). That is, higher within-system strength was associated with greater relative expression of the subgraph. This result implies that the subgraphs are also numbered roughly in order of their average relative expression.

(paragraph 3 of “Properties of subgraphs” on p.18)

Next, we investigated the relationship between these connectivity patterns and the temporal expression of each subgraph. As the matrix of functional connectivity was divided in two, with the first half reflecting positive connectivity and the second half reflecting negative connectivity, the temporal expression matrix also had two halves,

with the first reflecting positive expression over time and the second reflecting negative expression over time. As there was a strong negative correlation between positive and negative expression, we did all of our analyses on the relative expression (positive expression minus negative expression) of each subgraph²⁶. Across subgraphs, we calculated Pearson correlation coefficients between the average relative expression and the average within-system strength, average between-system strength, and average relative strength of each subgraph. To determine the significance of the correlation coefficients, we implemented 10,000 permutations of the subgraph labels to form the null distribution of correlation coefficients.

Comment 6:

I felt that the narrative and the scope of the main claims got lost in this version of the manuscript and I recommend seriously considering a careful re-reading and re-writing to improve clarity of the claims.

Response:

We thank the reviewer for this suggestion and have attempted a thorough re-writing, including re-organizing the discussion to make it more closely link to the introduction and results and to more clearly emphasize the main claims. We hope the main claims are much clearer in the revised manuscript.

Reviewer #3 (Remarks to the Author):

I enjoyed reading this manuscript reporting an analysis of how patterns of connectivity between brain regions change in tandem with two factors (sometimes referred to as surprise and uncertainty) that ought, normatively, to determine the speed of learning. The authors used an interesting way to identify patterns of interregional activity correlation that will be novel to many but which is explained very clearly and succinctly. I have suggested a few other pieces of information that the authors might provide to help readers understand their approach (point 1). Variation in activity in this inter-regional network (or subgraph) is then linked to variation in the factors determining learning across the task and variation across subjects. Final analyses confirm that these effects are not just attributable to interactions within the well-known “resting state” networks that have been defined previously or to univariate brain activity changes. Although my comments are prolix they are minor.

Response:

We thank the reviewer for this positive feedback.

Comment 1:

P4 The description of the method used to identify patterns of interregional activity correlation is explained in the last paragraph on page 4. Figures 2 and 3 are very helpful and the method is admirably clearly but also succinctly explained up to the sentence “The full data matrix was divided into two halves, with one half containing the positive correlation coefficients (zero if the coefficient was negative) and one half containing the absolute values of negative correlation coefficients (zero if the coefficient was

positive)²⁴” but the next sentence “We applied NMF to this matrix to identify functional subgraphs and their expression over time”, which is critical and which recurs figure legends could be clarified further at least somewhere near this point in the main text. The only other information about what constitutes a subgraph is in the Introduction: “patterns of functional connectivity across the entire brain”. A few more words of explanation either at the bottom of page 4 in the Results or in the Introduction would help.

Presumably a subgraph consists of nodes with timeseries of activity that exceed some threshold of correlation or other measure of similarity? Are they based only on positive correlation values? Can a given subgraph be identified on the basis of both positive correlations and the magnitude of negative correlations? Presumably there is some method or threshold for determining when subgraph is significantly separate and different from another and therefore a meaningful entity? It seems that they are distinct entities to the functional systems that are summarized in figure 2a on the basis of the Power study. However, a clear statement to this effect outside the Methods is likely to help many readers.

Response:

In the results (see paragraph 1-3 of “NMF identified ten subgraphs, or patterns of whole-brain functional connectivity, that varied over time” on p.5-6), we now provide more detail to clarify what subgraphs are, how we identified subgraphs based on both the positive functional connectivity and negative functional connectivity together, how we determined the number of subgraphs to extract, and how subgraphs are distinct from the functional systems identified in the Power et al (2011) study.

(paragraph 1-3 of “NMF identified ten subgraphs, or patterns of whole-brain functional connectivity, that varied over time” on p.5-6)

We used NMF to decompose whole-brain functional connectivity over time into specific patterns of functional connectivity, called subgraphs, and quantified the expression of these patterns over time. To perform NMF, we first defined regions of interest (ROIs) based on a previously defined parcellation²⁷ (Fig. 2a) and extracted blood-oxygenation-level-dependent (BOLD) time series for each ROI (Fig. 2b). For every pair of ROIs, we calculated the Pearson correlation coefficient between the BOLD time series in 10-TR (25 s) time windows, offset by 2 TRs for each time step (and thus 80% overlap between consecutive time windows). This procedure thus yielded a matrix whose entries represented time-dependent changes in the strengths of these pairwise correlations in the brain during the task. We unfolded each time window from this correlation matrix (Fig. 2c) into a one-column vector, and then concatenated these vectors from all time windows and all participants (Fig. 2d). As required for NMF, we transformed the resulting matrix to have strictly non-negative values: we duplicated the full matrix, set all negative values to zero in the first copy, and set all positive values to zero in the second copy before multiplying all remaining values by negative one. Thus, we divided the final full data matrix into two halves, with one half containing the positive correlation coefficients (zero if the coefficient was negative) and one half containing the absolute values of the negative correlation coefficients (zero if the coefficient was positive)²⁶. This procedure ensured that our approach did not give undue preference to

either positive or negative functional connectivity, and that subgraphs were identified based on both positive and negative functional connectivity.

We applied NMF to this matrix (A) to identify functional subgraphs and their expression over time. Specifically, we decomposed the full data matrix into a subgraph matrix W and an expression matrix H (Fig. 2d). The columns of W represent different subgraphs and the rows represent different edges (i.e., pairs of regions), with the value in each cell representing the strength of that edge (i.e., the functional connectivity strength for that pair of regions) for that subgraph. The rows of H represent different subgraphs, and the columns represent time windows, with the value in each cell representing the degree of expression of that subgraph in that time window. We implemented NMF by minimizing the residual error ($\|A - WH\|_F^2$) via three parameters: (i) the number of subgraphs (k), (ii) the subgraph regularization (α), and (iii) the expression sparsity (β) (Supplementary Fig. 1). The final result of NMF is a set of subgraphs (the number of which was determined by the parameter k), which reflected patterns of functional connectivity strengths across every pair of regions in the brain, as well as the expression of these subgraphs over time.

Using NMF, we identified ten subgraphs in our data. The full description of each subgraph specifies the edge strength between every pair of ROIs, corresponding to a 247x247 matrix. We calculated a simpler summary description that specifies the edge strength between every pair of functional systems in the previously defined parcellation, corresponding to a 13x13 matrix²⁷. Edges between ROIs were categorized according to the functional system of each ROI. To estimate the diagonal entries in the system-by-system matrix, we averaged the weights of all edges connecting two ROIs within a given system (Fig. 3a). To estimate the off-diagonal entries of the system-by-system matrix, we averaged the weights of all edges linking an ROI in one system with an ROI in another system. In line with common parlance, we refer to the edges within the same system as within-system edges, whereas we refer to the edges between two different systems as between-system edges. For presentation, we ordered and numbered the ten subgraphs according to the strength of within-system edges relative to that of between-system edges (Fig. 3b, Supplementary Fig. 2a-c). Finally, we thresholded the system-by-system matrix to show only edges that passed a permutation test ($p < 0.05$ after the Bonferroni correction for multiple comparisons; see Methods).

Comment 2:

Figure 4b illustrates the subgraphs – the correlated patterns of activity across regions – that were identified in a series of lateral views. Given the importance of subgraph 4 – the subgraph that seems most closely related to the variation in updating – it would be useful to also have a medial view illustration of subgraph 4. It might be helpful to clarify where some of the activity is found. For example, the subgraph is described as being frontoparietal which for some people will seem a very broad term given the limited number of lobes that the brain consists of. For others it may conjure up an image of the posterior lateral prefrontal cortex and intraparietal region. However, such an impression would not be quite correct; it looks as if it might be, within parietal cortex, predominantly the inferior parietal lobule that contributes to subgraph 4 and it looks as if the cingulate and frontal opercular regions are as important as lateral prefrontal cortex.

Response:

We thank the reviewer for this suggestion. We now summarize the components of subgraph 4 in two ways. In Fig. 4e, we illustrate all nodes from the four systems that have the strongest connectivity strength in subgraph 4. This figure shows the anatomical location of these four most important systems. Both medial and lateral views are provided. In Fig. 6c, we illustrate the individual nodes that are in the top 50% of connectivity strength, regardless of functional system. This figure shows the anatomical location of the regions with strongest connectivity strength in the subgraph and how these regions relate to the univariate results described in our previous study (McGuire et al., 2014). Again, both medial and lateral views are provided. We think that both methods for illustrating the subgraph are informative, and both provide a similar picture regarding the anatomical regions involved in subgraph 4.

Comment 3:

End of paragraph 1. “On the neural level, many studies have shown that such uncertainty and surprise in a dynamic environment was represented as univariate and multivariate activity in medial and lateral fronto-parietal networks1, 4, 5, 6.” Most of the studies cited here involve univariate fMRI analyses. Perhaps the recent multivariate analysis by Meder and colleagues (Nature Communications, 2017) might be cited here? The study seems relevant because, while not detracting from the novelty of the current manuscript, it also reports changes in correlation between multivariate patterns in four areas that appear to be have some overlap with those identified with subgraph 4 (although this could be clarified – see point 2).

Response:

Thank you for bringing this study to our attention. We now cite Meder *et al.* (Nature Communications, 2017) in the introduction.

Comment 4:

Typos

“participants’ predictions were influence by...”

Response:

Thank you for this. We have corrected the typographical error in the results (see the first sentence of paragraph 3 in the subsection entitled “Belief updating is influenced by normative factors related to uncertainty and surprise” on p.5.)

Other correction:

1. In the revised manuscript, we have also corrected Fig. 6. In the last version of Fig. 6, we found that data from left and right hemispheres were wrongly assigned to right and left hemispheres, respectively. This inaccurate lateralization affected the correlation coefficients for CPP (Fig. 6a) and RU (Fig. 6b) and the mapping of connectivity in subgraph 4 (Fig. 6c). After the correction, both the correlation coefficients for CPP (Fig.

6a) and RU (Fig. 6b) were enhanced. The mapping of connectivity in subgraph 4 (Fig. 6c) is now correct.

Reviewers' Comments:

Reviewer #1:

Remarks to the Author:

The authors have revised the manuscript significantly, thereby increasing the readability and clarifying my questions. I have no further comments or suggestions.

Reviewer #2:

Remarks to the Author:

Notes on resubmission of Kao-Kable Functional brain network reconfiguration during learning in dynamic environment for Nature Comm. Original Submission 8-19, resubmission 02-20.

The authors have made significant revisions and improvements to the original manuscript and responded to all prior critiques.

The result is mostly satisfying, however there are there are two small issues that nag.

1. The "CPP term" and "RU term" in the belief-update regression.

I still don't understand why the authors chose the specification that they did:

a) why no main effects for COO and RU?

b) why is the beta_3 specified with a (1-CPP) term?

2. What is the reference for the derivation of equation 5? The references seem to telescope ...

Reviewer #3:

Remarks to the Author:

The authors have dealt with the points that I have raised.

REVIEWERS' COMMENTS:

Reviewer #1 (Remarks to the Author):

The authors have revised the manuscript significantly, thereby increasing the readability and clarifying my questions. I have no further comments or suggestions.

Response:

We thank the reviewer for this positive feedback.

Reviewer #2 (Remarks to the Author):

Notes on resubmission of Kao-Kable Functional brain network reconfiguration during learning in dynamic environment for Nature Comm. Original Submission 8-19, resubmission 02-20.

The authors have made significant revisions and improvements to the original manuscript and responded to all prior critiques.

The result is mostly satisfying, however there are there are two small issues that nag.

1. The “CPP term” and “RU term” in the belief-update regression.

I still don't understand why the authors chose the specification that they did:

a) why no main effects for COO and RU?

b) why is the beta_3 specified with a (1-CPP) term?

Response:

In the belief-update regression analysis, our goal is to understand how different factors affect trial-by-trial learning rates, which is update in proportion to the prediction error ($\frac{B_{t+1}-B_t}{\delta_t}$). However, we perform a regression on updates ($B_{t+1} - B_t$) rather than on learning rates directly, as the errors in the former better conform to the assumptions of linear regression. Thus, the factors that we believe should affect learning rates appear in the regression as an interaction with prediction error:

$$\text{Update}_t = \beta_0 + \beta_1\delta_t + \beta_2\delta_t\Omega_t + \beta_3\delta_t(1 - \Omega_t)\tau_t + \beta_4\delta_t\text{Reward}_t + \beta_5\text{Edge}_t + \varepsilon \quad (6)$$

Note that updates and prediction errors are both signed, with positive (negative) values corresponding to rightward (leftward) errors and updates. β_0 then captures a bias to update in a rightward or leftward direction. As we do not expect that changes in CPP, RU or reward would lead to changes in updating in a rightward or leftward direction, we do not include these terms as “main effects”. If subjects use a fixed learning rate, β_1 will capture that learning rate. If instead subjects behave in accordance with the normative model, β_1 will be zero and β_2 and β_3 will be one. The specification of the CPP and RU terms is to keep this correspondence, given how learning rates are determined in the normative model (specifically, $\alpha_t = \text{CPP} + (1 - \text{CPP})\text{RU}$). Note that the learning rate, CPP and RU are all constrained to be between zero and one, and that the learning rate increases when either CPP or RU is high.

We also added detailed description under equation 3 (p. 13).

The learning rate, CPP and RU are all constrained to be between zero and one, and the learning rate increases when either CPP or RU is high.

We also added detailed description under equation 6 (p.14).

If subjects used a fixed learning rate (Eq. 2 alone), β_2 and β_3 will be zero and β_1 will reflect that fixed learning rate. In contrast, if subjects behave exactly in accordance with the normative model (Eq. 3), β_2 and β_3 will be one, and β_1 will be zero. Thus, we constructed the regression model so that the weights on β_2 and β_3 reflect the degree to which the two normative factors, CPP and RU, drive dynamic learning rates.

2. What is the reference for the derivation of equation 5? The references seem to telescope ...

Response:

Equation 5 (RU) was derived in Nassar et al. (2012, Nature Neuroscience). Unfortunately, though, this equation had a few typos in the originally published version of that paper, but was corrected in a later version to the following:

$$\tau_{t+1} = \frac{\Omega_t\sigma_N^2 + (1 - \Omega_t)\tau_t\sigma_N^2 + \Omega_t(1 - \Omega_t)(X_t\tau_t + B_t(1 - \tau_t) - X_t)^2}{\Omega_t\sigma_N^2 + (1 - \Omega_t)\tau_t\sigma_N^2 + \Omega_t(1 - \Omega_t)(X_t\tau_t + B_t(1 - \tau_t) - X_t)^2 + \sigma_N^2}$$

Note that, since $X_t - B_t$ is equivalent to the prediction error, δ_t , we rewrite the third term in the numerator and denominator and the equation appears as the following in our paper:

$$\tau_{t+1} = \frac{\Omega_t \sigma_N^2 + (1 - \Omega_t) \tau_t \sigma_N^2 + \Omega_t (1 - \Omega_t) (\delta_t (1 - \tau_t))^2}{\Omega_t \sigma_N^2 + (1 - \Omega_t) \tau_t \sigma_N^2 + \Omega_t (1 - \Omega_t) (\delta_t (1 - \tau_t))^2 + \sigma_N^2}$$

This is also exactly how the equation appears in McGuire et al. (2014, Neuron). Thus, to avoid any confusion, especially given earlier typos for this equation, we have just cited McGuire et al. (2014, Neuron) as the reference for equation 5.

Reviewer #3 (Remarks to the Author):

The authors have dealt with the points that I have raised.

Response:

We thank the reviewer for this positive feedback.